# The ozone–climate penalty over South America and Africa by 2100

Flossie Brown[1], Gerd A. Folberth[2], Stephen Sitch[3], Susanne Bauer[4,5], Marijn Bauters[6], Pascal Boeckx[6], Alexander W. Cheesman[3,7], Makoto Deushi[8], Inês Dos Santos[6], Corinne Galy-Lacaux[9], James Haywood[1,2], James Keeble[10,11], Lina M. Mercado[3,12], Fiona M. O'Connor[2], Naga Oshima[8], Kostas Tsigaridis[4,5], Hans Verbeeck[6]

[1] College of Engineering, Mathematics and Physical Sciences, University of Exeter, Exeter, UK
[2] UK Met Office Hadley Centre, Exeter, UK
[3] College of Life and Environmental Sciences, University of Exeter, Exeter, UK
[4] Center for Climate Systems Research, Columbia University, New York, NY, USA
[5] NASA Goddard Institute for Space Studies, New York, NY, USA
[6] Department of Environment, Ghent University, Ghent, Belgium
[7] Centre for Tropical Environmental and Sustainability Science, James Cook University, Australia
[8] Meteorological Research Institute, Japan Meteorological Agency, Tsukuba, Ibaraki, Japan
[9] Laboratoire d'Aerologie, Université Toulouse III Paul Sabatier, CNRS, Toulouse, France
[10] Yusuf Hamied Department of Chemistry, University of Cambridge, Cambridge, UK
[11] National Centre for Atmospheric Science (NCAS), University of Cambridge, UK
[12] UK Centre for Ecology and Hydrology, Wallingford, UK

*Correspondence to*: Flossie Brown (fb428@exeter.ac.uk)

## Abstract

Climate change has the potential to increase surface ozone ($O_3$) concentrations, known as the 'ozone–climate penalty', through changes to atmospheric chemistry, transport and dry deposition. In the tropics, the response of surface $O_3$ to changing climate is relatively understudied, but has important consequences for air pollution, human and ecosystem health. In this study, we evaluate the change in surface $O_3$ due to climate change over South America and Africa using three state-of-the-art Earth system models that follow the Shared Socioeconomic Pathway 3-7.0 emissions scenario from CMIP6. In order to quantify changes due to climate change alone, we evaluate the difference between simulations including climate change and simulations with a fixed present-day climate. We find that by 2100, models predict an ozone–climate penalty in areas where $O_3$ is already predicted to be high due to the impacts of precursor emissions, namely urban and biomass burning areas, although on average, models predict a decrease in surface $O_3$ due to climate change. We identify a small but robust positive trend in annual mean surface $O_3$ over polluted areas. Additionally, during biomass burning seasons, seasonal mean $O_3$ concentrations increase by 15 ppb (model range 12 to 18 ppb) in areas with substantial biomass burning such as the arc of deforestation in the Amazon. The ozone–climate penalty in polluted areas is shown to be driven by an increased rate of $O_3$ chemical production, which is strongly influenced by NOx concentrations and is therefore specific to the emissions pathway chosen. Multiple linear regression finds the change in NOx concentration to be a strong predictor of the change in $O_3$ production whereas increased isoprene emission

rate is positively correlated with increased $O_3$ destruction, suggesting NOx-limited conditions over the majority of tropical Africa and South America. However, models disagree on the role of climate change in remote, low-NOx regions, partly because of significant differences in NOx concentrations produced by each model. We also find that the magnitude and location of the ozone–climate penalty in the Congo basin has greater inter-model variation than in the Amazon, so further model development and validation is needed to constrain the response in central Africa. We conclude that if the climate were to change according to the emissions scenario used here, models predict that forested areas in biomass burning locations and urban populations will be at increasing risk of high $O_3$ exposure, irrespective of any direct impacts on $O_3$ via the prescribed emissions scenario.

## 1. Introduction

Climate change threatens to bring new pressures to the tropical forests, grasslands and agricultural lands of Africa and South America. As a result of shifts in emissions, atmospheric chemistry, and meteorology, as well as changes in vegetation behaviour such as transpiration rate, surface $O_3$ concentrations are likely to change (e.g. Turnock et al., 2019; Griffiths et al., 2021; Zanis et al., 2022), which may impair or benefit human and vegetation health (Agathokleous et al., 2019; Emberson, 2020) depending on the direction of change. $O_3$ is a highly oxidising compound, formed in the atmosphere through reaction of volatile organic compounds (VOC) or carbon monoxide (CO) with hydroxyl radicals (OH), and nitrate radicals ($NO_3$) in the presence of nitrogen oxides (NOx). However, it can also be removed from the atmosphere through reactions with many of the same chemical species (NOx, VOC, OH) depending on their relative concentrations. In addition to chemical pathways, $O_3$ can be removed from the lower atmosphere by dry deposition, which includes stomatal uptake by plants (Silva & Heald, 2018). Stomatal uptake of $O_3$ and subsequent ozone–plant damage, can lead to reduced carbon drawdown from the atmosphere (Sitch et al., 2007; Yue & Unger, 2018; Franz & Zaehle, 2020), and changes to biosphere–climate interactions (Sadiq et al., 2017). Sitch et al. (2007) showed that the tropics may be especially sensitive to high $O_3$ concentrations and therefore susceptible to large productivity losses if surface $O_3$ were to increase. Additionally, $O_3$ is a near-term climate forcer with impacts on the radiative balance leading to a positive radiative forcing of climate through absorption of longwave radiation (Myhre et al., 2017). As the tropical forests are vital as sinks for atmospheric $CO_2$, and tropical ecosystems play a vital role in regional and global climate (Lewis, 2006; Bonan, 2008), an understanding of the impact of climate change on surface $O_3$ concentrations in the tropics is critical.

Whilst there have been no studies specifically assessing changes in surface $O_3$ due to climate change in the tropics, global studies have suggested that chemical and biological changes in temperature-dependent chemistry, natural emissions of precursors and land surface properties; as well as dynamical changes including circulation changes and transport from the stratosphere may lead to an 'ozone–climate penalty' over some continental areas (Jacob & Winner, 2009; Doherty et al., 2013;

Zanis et al., 2022). The 'ozone–climate penalty' is defined as an increase in surface $O_3$ concentrations due to climate change alone. It is influenced by many complex chemical and biological processes, which are not all well-understood or represented in current climate models, although there has been substantial recent research to reduce uncertainty in temperature-sensitive chemistry, meteorology and land–atmosphere feedbacks (Oswald et al., 2015; Coates et al., 2016; Sadiq et al., 2017; Romer et al., 2018; Archibald et al., 2020b). There has been relatively little research focusing on $O_3$ in tropical environments. Unlike the more commonly studied extra-tropical Northern hemisphere, the tropics and subtropics have relatively low (natural) NOx emissions and high biogenic VOC emissions, high actinic flux and strong atmospheric convection (Bond et al., 2002; Pugh et al., 2010; Paulot et al., 2012). This paper focuses on South America and Africa. We exclude equatorial Asia because the atmospheric chemistry in this region is more uncertain due to difficulties in detecting and accounting for peat fire emissions (Prosperi et al., 2020). Equatorial Asia also has a greater marine influence with most model grid boxes containing ocean as well as land, so they may follow a different chemical regime.

Many areas in Africa and South America are considered remote (defined in this paper by low emissions of NOx), although increasing anthropogenic activity such as urbanisation and biomass burning causes moderate NOx emissions in some regions (e.g. Kuhn et al., 2010; Pacifico et al., 2015; Shi et al., 2020). The sensitivity of $O_3$ production to NOx depends on the relative concentrations of NOx and VOCs. Isoprene is the most abundant biogenic VOC in remote Africa and South America and must be oxidised in the atmosphere before it can form $O_3$ (Liu et al., 2016). NOx is produced from both natural and anthropogenic sources including soils, lightning, transport and biomass burning. In NOx-limited regimes where $O_3$ production is proportional to NOx concentrations, increasing VOCs or OH can also act toreduce $O_3$ concentrations through oxidation and formation of organic peroxides (Pacifico et al., 2012). In this NOx-limited case, increasing NOx will lead to greater $O_3$ formation. Conversely, in VOC-limited regions with sufficient NOx present, increasing NOx concentrations may reduce $O_3$ concentrations by removal of the key $O_3$-forming radicals OH (reaction: $OH + NO_2 \rightarrow HNO_3$).

Earlier studies have found that South America and Africa are generally NOx-limited (Ziemke et al., 2009; Bela et al., 2015), and that increases in NOx concentration associated with climate change will be a key driver of $O_3$ increases over South America and Africa. Doherty et al. (2013) attribute the NOx increase predominantly to enhanced decomposition of the NOx reservoir species, peroxyacetyl nitrate (PAN). A fraction of emitted NOx is locked up as PAN, which decomposes back into NOx in warmer temperatures, sometimes after having travelled long distances from the NOx source. As PAN is unstable at high temperatures, climate change will result in a smaller fraction of NOx being stored as PAN in NOx source regions and may also decrease the amount of NOx that is transported into remote regions (Schultz et al., 1999; Finney et al., 2018). Lightning NOx is known to contribute to $O_3$ formation, however studies project both increases and decreases in future lightning frequency (Clark et al., 2017; Finney et al., 2018) leading to low confidence in how $O_3$ will be affected by climate-driven changes in lightning (Fu & Tian, 2019; Murray, 2016).

The role of isoprene in the ozone–climate penalty is debated as there is uncertainty about how isoprene emissions will change in the future in response to temperature, $CO_2$ and land-use change (Fu & Liao, 2016; Fu & Tian, 2019) and how to best represent isoprene chemistry in climate models (Weber et al., 2021). Biogenic isoprene emissions increase strongly with temperature and vegetation stress (e.g. Guenther et al., 1993; Niinemets et al., 1999; Unger et al., 2013; Morfopoulos et al., 2021), but very high temperatures or moisture stress may cause 'die-back' of vegetated areas, which would decrease isoprene emissions overall (Sanderson et al., 2003; Cox et al., 2004; Malhi et al., 2009). On the other hand, elevated $CO_2$ concentrations directly inhibit isoprene emission but can indirectly increase emission if this $CO_2$ fertilisation effect results in increased plant productivity (Pacifico et al., 2011; Squire et al., 2014; Hantson et al., 2017). Isoprene, NOx and OH concentrations are influenced by isoprene chemistry. Formation of isoprene nitrates partially recycles NOx, and oxidation of isoprene partially recycles HOx (Bates and Jacob, 2019). Difference between models in their calculation of the yields and recycling rates of these species is likely to affect $O_3$ concentrations.

Besides chemical processes, dry deposition of $O_3$ to vegetation is a major $O_3$ sink, accounting for 20% of $O_3$ removal (Wedow et al., 2021). Most $O_3$ deposition occurs via plant stomata, which respond to changes in the climate. (Silva & Heald, 2018; Clifton et al., 2020). Increased $CO_2$, temperature and vapour pressure deficit (VPD) will decrease stomatal conductance and therefore decrease $O_3$ deposition rate. This could lead to increased concentrations of $O_3$ in the atmosphere, although it would have a protective effect for plants (Emberson et al., 2013; Lin et al., 2020).

Studies agree that over the ocean, average surface $O_3$ concentrations will decrease under the influence of climate change (Zeng et al., 2008; Doherty et al., 2013; Zanis et al., 2022). The warmer air can hold more water vapour, a major species contributing to $O_3$ loss (reaction: $O(1D) + H_2O \rightarrow 2OH$ leading to $O_3$ loss via $OH_2 + O_3 \rightarrow OH + 2O_2$). There may also be $O_3$ reductions in remote regions over land due to this process although natural emissions can change the atmospheric chemistry over the continents. Other factors contributing to the $O_3$ concentration over the continents are numerous and complex, including changes to oxidising capacity, stratospheric transport and land-use (Archibald et al., 2010; Squire et al., 2015).

This paper quantifies the effect of climate change on surface $O_3$ concentrations in the future, and aims to understand its uncertainty and the relative contributions of the underlying processes. We focus on areas with robust $O_3$ changes to identify areas in South America and Africa at greatest risk by 2100. Sect. 1 has introduced the key chemical species involved in $O_3$ chemistry and the most important changes that may result from climate change. In Sect. 2, we provide the model details, data used for evaluation, and description of analysis of model output. Sect. 3 evaluates model predictions of surface $O_3$ in the present-day, evaluates model predictions for surface $O_3$ changes in 2100 and examines the importance of chemical and deposition changes in controlling the ozone–climate penalty in models. Finally, Sect. 4 discusses the key trends predicted by the models, limitations of the study and crucial uncertainties in the models.

## 2. Methods

We analyse surface $O_3$ for simulations that follow a medium-high emissions pathway created for CMIP6 (Pascoe et al., 2019). The simulations were carried out as part of an ensemble of Earth system model experiments designed to quantify the climate and air quality impacts of aerosols and trace gases in climate models (Collins et al., 2017), named Aerosol-Chemistry Model

Intercomparison Project (AerChemMIP). For this study three Earth system models were used: UKESM1-0-LL (abbr. UKESM1), GISS-E2-1-G (abbr. GISS), MRI-ESM2-0 (abbr. MRI). These were selected because of their detailed tropospheric chemistry schemes and availability of output for $O_3$ and $O_3$ precursor concentrations on the Earth System Grid Federation (ESGF).

The simulations follow the Shared Socioeconomic Pathway 3-7.0 (SSP3-7.0) emissions scenario, a scenario assuming low international cooperation to protect the environment. This includes high emissions of non-methane near-term climate forcers and substantial land-use change (O'Neill et al., 2016; Gidden et al., 2019). The prescribed emissions include anthropogenic and biomass burning emissions of NO, $CO_2$ and CO (Rao et al. 2017; Riahi et al., 2017) and the future pathway for $CH_4$ is calculated as an atmospheric concentration (Meinshausen et al., 2019). Emissions due to growing populations and poor

international cooperation results in significant temperature increases by 2100 (Turnock et al., 2019) and societies that are highly vulnerable to climate change. This emissions pathway was chosen in order to understand changes in end-of-century $O_3$ concentration if there is no international cooperation to reduce precursor emissions.

### 2.1 Model descriptions

A comparison among models of natural emissions that may respond to the climate are shown below (Table 1). Where emissions

are prescribed, the source is provided. Emissions that are interactive will respond to climate change. Further details on each of the Earth system models, including descriptions of the interactive emissions schemes and the tropospheric chemistry schemes are provided below.

|  | Isoprene | Terpenes | Other VOCs | Soil NOx | Lightning NOx |
|---|---|---|---|---|---|
| UKESM1 | Interactive (Pacifico et al., 2011) | Interactive (Guenther , 1995) | MACCity-MEGAN (Sindelarova et al., 2014) | Yienger & Levy (1995) | Interactive (Price & Rind, 1992) |

| GISS | Interactive (Guenther , 1995) | Lathiere et al. (2005) | Lathiere et al. (2005) | GEIA (Guenther et al., 1995), | Interactive (Price & Rind, 1992) |
| MRI | GEIA (Guenther et al., 1995), | GEIA (Guenther et al., 1995), | Müller et. al. (1992) | Yienger & Levy (1995) | Interactive (Price & Rind, 1992) |

**Table 1: Sources of natural emissions of ozone precursors. Where emissions are prescribed, the source is provided. Interactive emissions respond to climate change.**

**UKESM1-0-LL (abbr. UKESM1)**

UKESM1-0-LL is a combination of HadGEM3 (Williams et al., 2018) with additional land and atmospheric chemistry components (Sellar et al., 2019). The UK Chemistry and Aerosol scheme (UKCA) contains stratospheric and tropospheric chemistry (Archibald et al., 2020a) combined with the GLOMAP-mode aerosol microphysics scheme (Mulcahy et al., 2018, Mulcahy at al., 2020).

Interactive emissions include isoprene, monoterpenes and lightning NOx. Isoprene and monoterpene emissions respond to light and temperature (Archibald et al., 2020a; Mulcahy et al., 2018). Isoprene emissions are calculated from vegetation productivity and increase in response to light and temperature (with an optimum at 40 °C). Emissions of isoprene are inhibited by CO2 following the emission model of Pacifico et al. (2011). Lightning NOx is calculated using the parameterisation of Price and Rind (1992) , which calculates a lightning flash density based on cloud-top height. Nitrogen oxide molecules produced per flash is 7.5 x1026 for cloud-to-ground flashes and 2.25 x1026 for cloud-to-cloud flashes. Secondary organic aerosols (SOA) are calculated as a fixed yield of 26% from gas-phase oxidation reactions involving monoterpene sources. Soil NOx is prescribed as an annual flux of 12 Tg, according to Yienger and Levy (1995) and other biogenic emissions are prescribed as monthly mean climatologies based on the years 2001–2010 (Guenther et al., 2012).

The terrestrial biogeochemistry is provided by JULES (Wiltshire et al., 2019; Wiltshire et al., 2021). Stomatal conductance in JULES is similar to the Ball-Berry-Leuning model (Leuning, 1995) and responds to the ratio of internal to external $CO_2$ concentrations and leaf humidity deficit (Jacobs, 1994).

The model has a horizontal resolution of 1.25° latitude by 1.875° longitude with 85 vertical levels in a hybrid height coordinate.

**MRI-ESM2-0 (abbr. MRI)**

MRI-ESM2-0 (Yukimoto et al., 2019; Kawai et al., 2019; Oshima et al., 2020) contains an atmospheric and land-surface model
(MRI-AGCM3.5), an ocean and sea-ice model (MRI.COMv4), an aerosol model (MASINGAR mk-2r4c) and an atmospheric
chemistry model (MRI-CCM2.1). The chemistry model includes tropospheric, stratospheric and mesospheric chemistry with
90 chemical species and 259 chemical reactions (Deushi & Shibata, 2011).

Lightning NOx is interactive and based on a lightning flash density parameterisation (Price & Rind, 1992). A cloud-to-ground
flash produces $6.7 \times 10^{26}$ molecules per flash and a cloud-to-cloud flash produces $6.7 \times 10^{25}$ molecules per flash. Other natural
emissions from land and ocean are prescribed as monthly climatologies, including isoprene and soil NOx (Deushi & Shibata,
2011). 15 % of natural terpene emissions at the surface form SOA and SOA have identical properties to POA.

Each component employs different horizontal resolutions but the outputs used in this paper are from the chemistry component
which uses a horizontal resolution of 2.8125° latitude by 2.8125° longitude with 80 vertical levels in a hybrid sigma pressure
coordinate.

**GISS-E2-1-G (abbr. GISS)**

GISS-E2-1-G contains a coupled troposphere and stratosphere chemistry scheme using the G-PUCCINI chemistry mechanism
(Shindell et al., 2013; Kelley et al., 2020) combined with the One-Moment Aerosol (OMA) scheme for aerosols (Bauer et al.,
2020).

Lightning NOx is interactive as described by Kelley et al. (2020). The NO yield is $1.75 \times 10^{26}$ molecules per flash. Soil NOx
is prescribed as a monthly mean from GEIA. Isoprene emissions are interactive and which respond to light and temperature
(Shindell et al., 2006) following the algorithm defined by Guenther et al., (1995). Monoterpenes are prescribed as monthly
means from Lathiere et al. (2005) based on the year 1990. SOA are calculated using the CBM4 chemical mechanism to describe
the gas phase tropospheric chemistry together with all main aerosol components including SOA formation and nitrate, and is
calculated using four tracers in the model. Isoprene (VOCs) contribute to the formation of SOA (Tsigaridis et al., 2018).

GISS has a horizontal resolution of 2.00° latitude by 2.25° longitude with 40 vertical levels output on hybrid sigma pressure
coordinate.

## 2.2 Data analysis methods

### Model evaluation

In situ observations from 65 sites across South America and tropical Africa, covering key biomes and land-use types, are used for grid level model evaluation. Monthly mean $O_3$ concentrations from individual sites have been aggregated into seven regions by grouping together sites within latitude and longitude bounds (see Table S1). To compare to models, the coordinates of the in situ measurement sites are matched to the nearest model grid cell coordinates. To create an average seasonal cycle for each region, sites with the same nearest grid cell are averaged together to create a grid cell seasonal cycle. Then, grid cell seasonal cycles in the same region are averaged together. Monthly mean data from 1990 up to 2021 were used, although most sites only provide a few years of data. Data were excluded if there was an unequal distribution of data points over the monthly mean diel cycle.

To produce a model seasonal cycle, the monthly mean $O_3$ concentration was calculated using CMIP6 historical simulations for the period 1990–2014. This was done for each grid cell that contained an observation site. The standard deviation in monthly mean $O_3$ concentrations between the grid cells used was calculated for each region (i.e. it represents variation in $O_3$ geographically between the sites rather than inter-annual variation).

For model evaluation against Tropospheric Emission Spectrometer (TES) data from the Aura satellite, which retrieves $O_3$ in ppb over 67 pressure levels, we use $O_3$ concentrations at the lowest level available from TES (825 hPa). Monthly mean gridded outputs are used for the period 2004–2011, the period for which complete monthly mean data is available. As with the model output, satellite grid cell coordinates closest to the in situ site coordinates were selected. The monthly mean and standard deviation for each region was calculated, only using data from grid cells containing in situ sites.

### CMIP6 model output

We isolate the effect of climate change on surface $O_3$ concentrations using the difference between two simulations which consider the same trajectory of anthropogenic emissions changes but differ in climate. The simulations are global model runs driven with prescribed sea surface temperatures (SSTs) over the period 2015–2100. We use a simulation driven with changing SSTs from the coupled simulation ssp370 so that the climate changes in accordance with the emissions changes, this simulation is named ssp370SST. We use a second simulation with prescribed SSTs and sea ice concentrations taken from a present-day climatology (2005–2014) in historical simulations, named ssp370pdSST. Importantly, although anthropogenic emissions are identical in both simulations, ssp370pdSST does not include the resulting climate change.

To isolate the effect of climate change on tropospheric $O_3$, we subtract ssp370pdSST (present-day constant climate + future emissions) from ssp370SST (future climate + future emissions) following Zanis et al. (2022) and Szopa at al. (2021). Biomass

burning and land-use change are considered anthropogenic and are prescribed for both models but natural emissions are allowed to change depending on the model set-up. In this way, the background atmospheric composition is based on the future emissions scenario used, since the response of atmospheric chemistry to climate change may depend on the background concentrations of precursors. In UKESM1, $CO_2$ is also fixed to present-day concentrations in ssp370pdSST so that the effect of climate change includes the effect of $CO_2$ inhibition.

Model output is taken as monthly means during the period 2090–2100 between 40° S and 40° N from experiments ssp370SST and ssp370pdSST. All variables used are outlined in Table S2. The change due to climate change refers to subtracting ssp370pdSST from ssp370SST, where positive values are considered an $O_3$–climate penalty. When evaluating the change due to climate change regionally, this study distinguishes between 'high-NOx' and 'remote' areas (Fig. 1). High-NOx areas are defined as areas where the annual mean NOx emissions are above the 95[th] percentile for the tropics and subtropics (40° S–40° N).

$O_3$ concentrations are taken from the lowest model level (~ 20 m above orography). Where multi-model means are shown, data has been re-gridded to the 2.8125° by 2.8125° grid used by MRI. We evaluate the ozone–climate penalty as a yearly average and seasonally. To identify seasonal patterns we aggregate the data by burning season. The Western African burning season is defined as Dec–Feb, the Southern African burning season is June and July and the Southern Amazon burning season is Aug–Oct.

To attribute the ozone–climate penalty to precursor variables, we also use NOx, isoprene emission rate, OH and surface temperature variables. When presenting these variables, the ocean has been masked so that only land surface changes are presented. 'Surface concentrations' shown in this study refer to chemical mixing ratios in the lowest model grid cells and 'background concentrations' refer to chemical mixing ratios in the absence of climate change (using data from ssp370pdSST).

To evaluate the $O_3$ budget, we also use $O_3$ chemical production, $O_3$ chemical loss and dry deposition variables. $O_3$ chemical production is defined as $O_3$ produced from the reaction $NO+RO_2$ / $HO_2$, $O_3$ chemical loss is the sum of (i) $O(1D)+H_2O$; (ii) $O_3+HO_2$; (iii) $O_3+OH$; (iv) $O_3+$alkenes (e.g. isoprene). To compare these three variables on the same scale, we convert the units to Tg year$^{-1}$ and sum production and loss over the lowest 1 km, the approximate boundary layer height. We choose 1 km to establish an approximate region that can contribute to surface $O_3$ concentrations.

In Sect. 3.4, we present a sensitivity study relating changes in NOx concentration and isoprene emissions to changes in $O_3$ chemical production. We use monthly mean data in South America and Africa within 30° S and 30° N to calculate a monthly climatology for 2090–2100, masking the ocean and the non-vegetated region of Saharan Africa. To identify the limiting $O_3$ precursor in the tropics and subtropics, the percentage change in $O_3$ production rate (n>500) is modelled with an ordinary least

squares linear regression using the percentage change in NOx and isoprene as predictor variables (see S4: The relationship between NOx and $O_3$ production). To interpret the ability of the model we rely on the central limit theorem to assume that the normalised sum of the residuals can be approximated by a normal distribution. Unique months and grid cells are treated as
separate data points. To present the results graphically, we highlight values using star markers if they are above a threshold of the 95[th] percentile for NOx concentrations using the monthly climatology for each model individually, with aims to identify biomass burning areas and cities. The atmospheric chemistry in these areas may be different due to the elevated NOx concentrations. Therefore, some grid cells will be above the 95[th] percentile for specific months only (biomass burning seasons).

## 3. Results

**3.1. Evaluation of model skill for present-day surface $O_3$ concentrations**

Results show that climate models are able to capture the observed seasonal cycle in most regions except for West Africa and DR Congo (Fig. 1). However, the models overpredict monthly mean surface $O_3$ concentrations by up to 50 ppb, with the largest bias present in remote forest locations such as the Congo area (Fig. 1e). GISS overall has the smallest positive bias out of all the models, and MRI has the largest. UKESM1 shows the smallest seasonal variation in $O_3$ concentration, which is often closer
to the observed seasonal pattern.

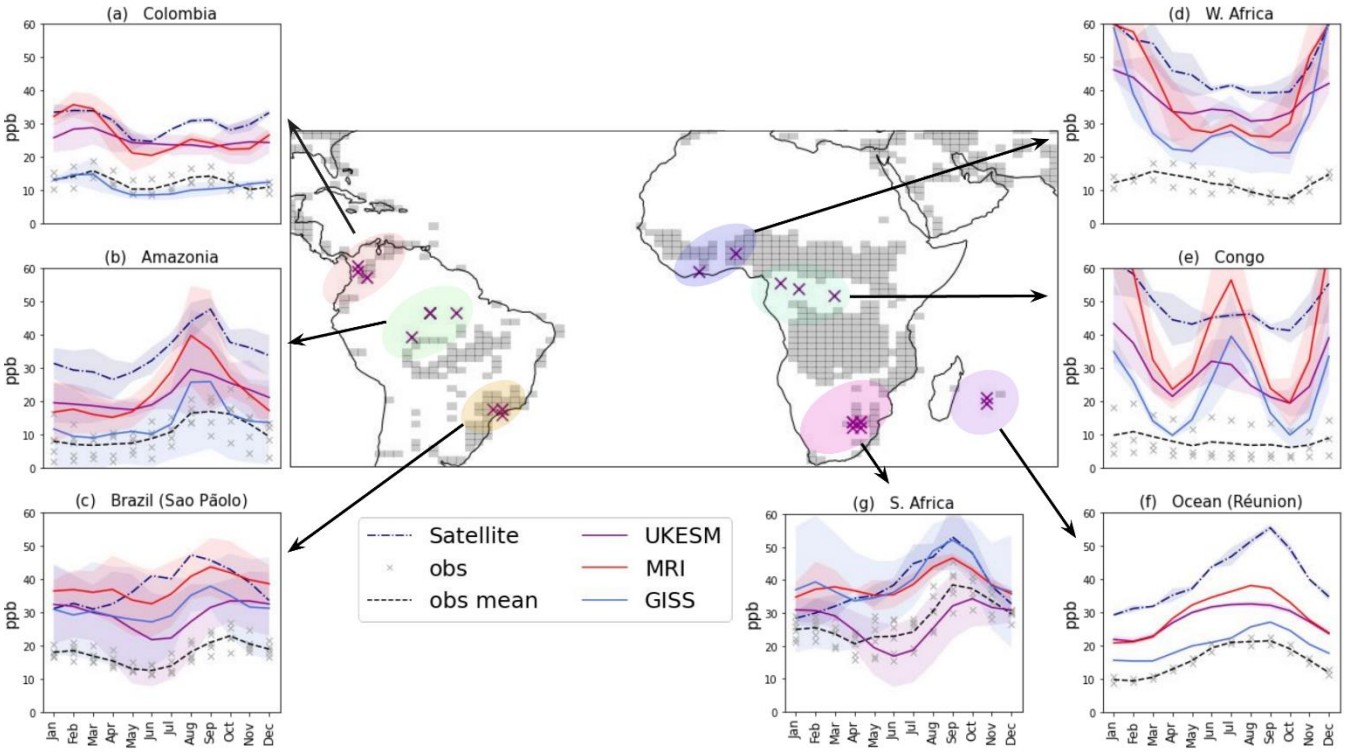

**Figure 1: (a)–(g)** compare monthly mean O₃ (black dashed line) from site measurements averaged over 7 regions to model predictions for the period 1990–2014 and satellite products from the TES satellite at 825 hPa (navy dash-dot line). Model means are shown for UKESM1 (purple solid line), GISS (blue solid line) and MRI (red solid line) and 2 standard deviations from the mean are shaded. Grey crosses indicate monthly mean O₃ measurements from individual sites and years.

The central panel marks the locations where O₃ measurement sites are located (purple crosses) and how the sites have been grouped (coloured shading). Observations (a)–(c) and (f)–(g) use data from the Tropospheric O₃ Assessment Report (TOAR I, Schröder, 2021; Schultz et al., 2017), (d) and (e) from INDAAF (http://www.indaaf.obs-mip.fr) and (e) from CONGOFLUX in Yangambi, DR Congo. Grey grid cells cover areas where NOx emissions are above the 95th percentile for the region 40° S–40° N in 2100 (prescribed by (Rao et al., 2017; Riahi et al., 2017), indicative of biomass burning areas and cities. Un-shaded areas are referred to as 'remote' throughout.

Grey shading on Fig. 1 highlights the areas in South America and Africa with the highest NOx emissions. The shaded areas represent areas with high biomass burning emissions and urban areas and are referred to as 'high-NOx' areas in this paper. In the Southern Amazon (Fig. 1b), the biomass burning months are August and September and both models and observations predict the highest O₃ concentrations in this season. However, the observed monthly mean O₃ concentrations range between 9 to 20 ppb whereas models predict values up to 40 ppb, with GISS displaying the smallest positive bias. In Africa (Figs 1d–1f), the biomass burning months are December–February (North / West Africa) moving to June–July (Southern Africa). Whilst models predict concentrations of up to 80 ppb in the Congo during these months due to transport of precursors from biomass burning, observations show substantially lower surface O₃ concentrations of less than 20 ppb at the remote locations sampled, although the highest O₃ concentrations occur during December–February (Adon et al., 2010; 2013; Ossohou et al., 2019). In fact, seasonal variation is low at the Congo sites whereas models predict strong seasonal patterns (Fig. 1e). In months without substantial burning, GISS captures the low O₃ values well and UKESM1 and MRI overestimate by 10 to 15 ppb.

The enhanced O₃ concentrations predicted by models in burning seasons over the Congo and West Africa are also captured in satellite retrievals at 825 hPa. Satellite O₃ concentrations in the DR Congo are 10 ppb higher in December–February compared to months without burning (Fig. 1e). At 825 hPa, these satellite capture O₃ concentrations above the altitude of in situ observation sites and the lowest model level. Model predictions for O₃ at 825 hPa are ~10 ppb higher than the lowest model level and therefore compare well to satellite retrievals (Fig. S1), especially in Amazonia (Fig. S1b) and West Africa (Fig. S1d).

Similar to biases over the remote continents, measurements from Réunion Island (Fig. 1f), which capture oceanic air masses, are overestimated by 5 ppb in GISS and by 12 ppb in UKESM1 and MRI, although the seasonal cycle is reproduced. On the other hand, over the urban sites in South Africa, Johannesburg, UKESM1 replicates the observed mean with moderate accuracy, whereas GISS and MRI overestimate by 15 ppb (Fig. 1g).

### 3.2. Average changes to atmospheric composition over Africa and South America at the end of 2100

The overall change in O₃ concentration due to climate change over the African and South American land surface is shown in Fig. 2a for each model compared to the simulation with a fixed present-day climate. All models predict that on average O₃

concentrations will decrease due to climate change over the tropical land surface in this study. The magnitude of the predicted

$O_3$ change for each model will depend on background concentrations of $O_3$ chemical precursors, their change due to climate change and individual model mechanistic details. There is significant diversity in background atmospheric composition between models (Fig. 2, blue marker) and the direction of change in surface NOx and OH concentration due to climate change (Figs 2b, 2c, arrows).

Different temperature sensitivities of the models (differing by up to 2.7 K) and different biogenic VOC schemes will contribute to the inter-model variation (Fig. S2). UKESM1 has the greatest temperature sensitivity with a 6.5 K increase in temperature over the tropical land surface due to climate change (Fig. 2e). The temperature change due to climate change varies seasonally and regionally, which may affect concentration of $O_3$ precursors locally, with dry seasons temperatures predicted to rise by 1–1.5 K more than wet season temperatures (Fig. S3).

The change in NOx concentration between models is determined by the balance of changes in isoprene nitrate formation, OH concentrations, PAN decomposition and lightning in the models. A decrease in NOx concentrations could be related to changes in OH concentration and precipitation (and thus NOx removal via reaction $NO_2 + OH \rightarrow HNO_3$) and isoprene (and thus NOx removal via isoprene nitrate formation), whereas NOx concentration increases may be due to increased PAN decomposition

or lightning.

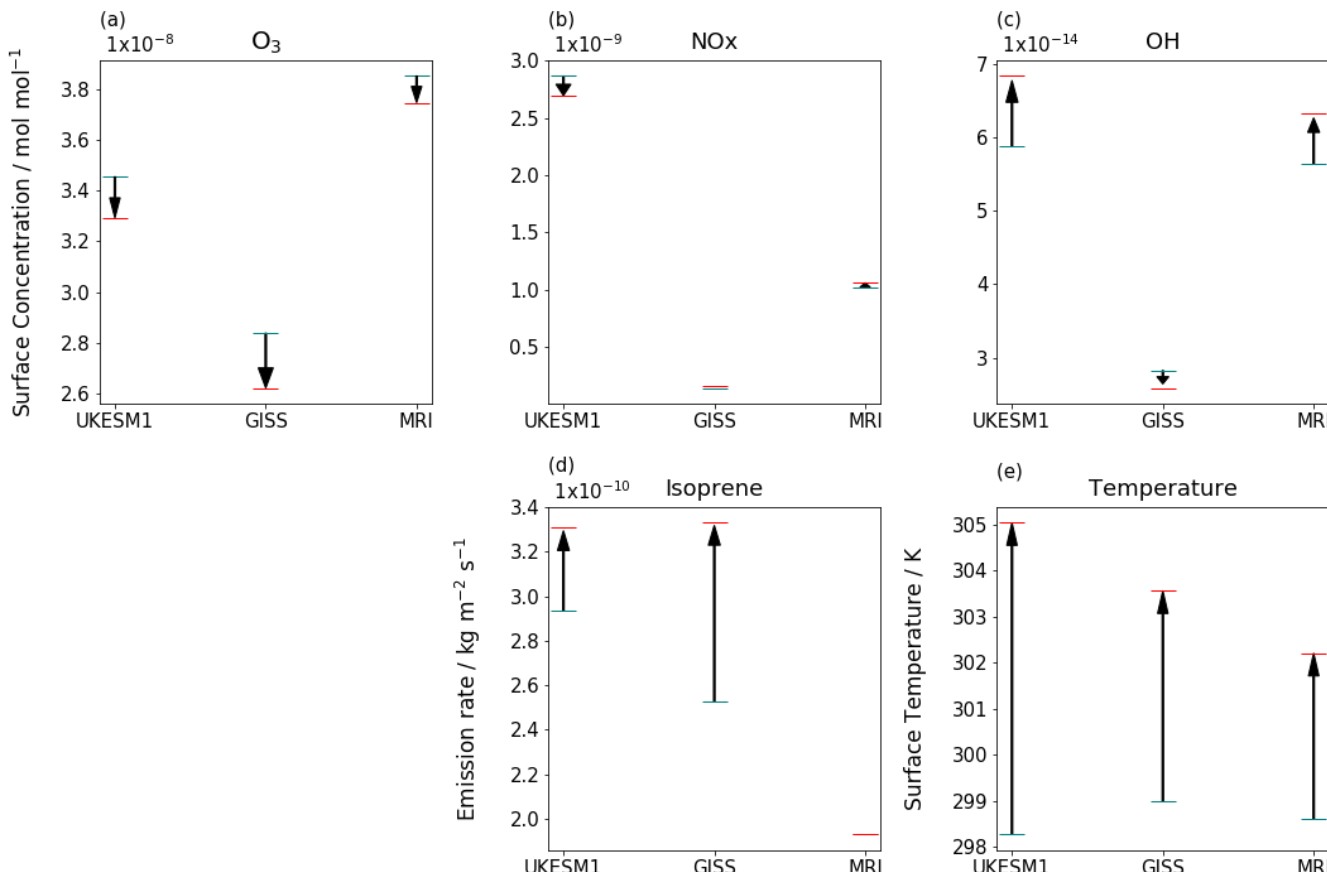

**Figure 2: The change in surface concentration of (a) O₃, (b) NOx, (c) OH, and the change in (d) isoprene emission rate and (e) surface temperature from experiment ssp370pdSST with no climate change (blue line) to experiment ssp370SST with climate change (red line) for the three climate models in this study. Variables have been averaged over the African and South American continents between 12° N–30° S for the period 2090–2100. The change due to climate change is significant at the 5% level for all variables and models except isoprene in MRI (which is prescribed so does not change).**

Anthropogenic NOx emissions, including biomass burning emissions, are prescribed based on the SSP3-7.0 scenario, soil NOx is prescribed by each model and lightning NOx differs between the models based on the chosen parameterisation of individual models. Compared to the present-day, NOx emissions in biomass burning areas decrease in Africa to follow projected trends, but do not change in South America. NOx emissions increase in cities and Nigeria especially has major growth in urban areas. Compared to the scenario without climate change, total lightning NOx emissions increase in all models, and the increases occur during the wet season (Fig. S4). MRI predicts much larger increases than GISS and UKESM1, and UKESM1 shows a decrease in lightning NOx over the Amazon basin in December–February (Fig. S4a) although the net effect over all seasons is positive. Peroxyacetyl nitrate (PAN) decreases in all models ($-94$ ppt, $-61$ ppt, $-30$ ppt for UKESM1, GISS and MRI respectively) due to increased thermal decomposition. In GISS and UKESM1, the increase in isoprene emissions can increase removal of NOx via formation of isoprene nitrates.

Hydroxyl radical (OH) concentration determines the oxidising capacity of the atmosphere and affects rates of reaction such as VOC oxidation and ozone destruction. Increased temperatures will increase atmospheric water vapour and OH production, however OH concentrations decrease in the GISS model when climate change is included. A portion of this decrease can be attributed to an increase in isoprene emissions, which is much larger in GISS than UKESM1 (Fig. S2).

The increase in isoprene emission rate due to climate change depends on the isoprene emission scheme used, or in MRI, isoprene emissions are prescribed as a climatology. The greatest increase in isoprene emissions rate occurs in the GISS model, which increases from $2.5 \times 10^{-10}$ kg m$^{-2}$ s$^{-1}$ to $3.3 \times 10^{-10}$ kg m$^{-2}$ s$^{-1}$ when climate change is considered, whereas UKESM1, which accounts for $CO_2$ inhibition, increases more modestly from 2.9 kg m$^{-2}$ s$^{-1}$ to $3.3 \times 10^{-10}$ kg m$^{-2}$ s$^{-1}$. Isoprene emissions are presented throughout rather than isoprene concentrations (see S2: Isoprene representation in this paper).

### 3.3 Changes in surface $O_3$ concentration due to climate change over remote regions compared to high-NOx areas

This study focuses on the change in surface $O_3$ concentration over land in 2100, although we note there are significant decreases in $O_3$ concentration over the oceans and non-vegetated areas such as Saharan Africa (Fig. 3). Over land, the multi-model mean shows increases of up to 4 ppb over urban areas and the biomass burning areas of South America and Africa (Fig. 3a) whereas ocean-influenced locations such as Northeast Brazil are expected to benefit by a 4 to 5 ppb decrease in surface $O_3$. However, the direction of change in surface $O_3$ concentration over central Africa and the remote Amazon (North West) is not robust between models (Fig. 3c, 3d, 3e).

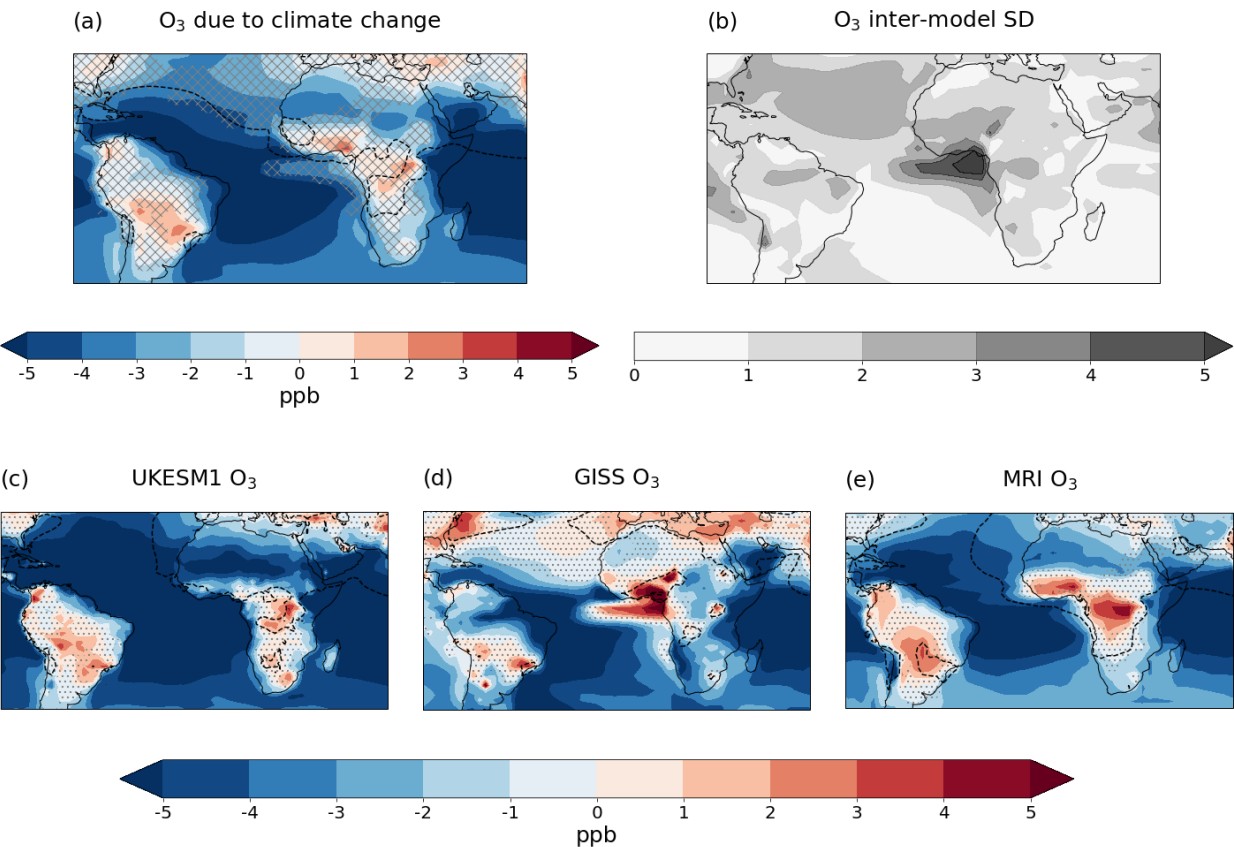

**Figure 3: The average change in surface O$_3$ concentration due to climate change for the period 2090–2100 for (a) the multimodel mean, (c) UKESM1 only, (d) GISS only, (e) MRI only. (b) shows the inter-model standard deviation in the same units. Grey hatching in (a) covers areas where the inter-model standard deviation is greater than 20% of the multimodel mean value. Grey dots in panels (c)–(e) cover areas that are not significant at the 5% level from a student's t-test. Black dotted lines outline areas where background O$_3$ is higher than 40 ppb.**

UKESM1 and MRI predict increases in surface O$_3$ concentration of up to 5 ppb in the Amazon and central Africa, with decreases over coastal regions due to climate change (Figs 3c, 3e). In the remote Amazon, MRI predicts an increase and UKESM1 a decrease in O$_3$ concentration, but neither change is significant. On the other hand, GISS predicts significant O$_3$ decreases across remote regions of up to 4 ppb (Fig. 3d), including central Africa, which experiences O$_3$ increases in the other simulations (Figs 3c, 3e).

Changes in surface O$_3$ concentration due to climate change in 2100 are shown in Fig. 4, grouped by regional biomass burning season, with dotted contours where background O$_3$ is 40 ppb (a number assumed associated with thresholds for plant O$_3$ damage) and 70 ppb. High background O$_3$ is associated with biomass burning and pollution in and around cities due to their higher NOx emissions. These high O$_3$ areas also show the greatest increase in O$_3$ due to climate change (Fig. 4).

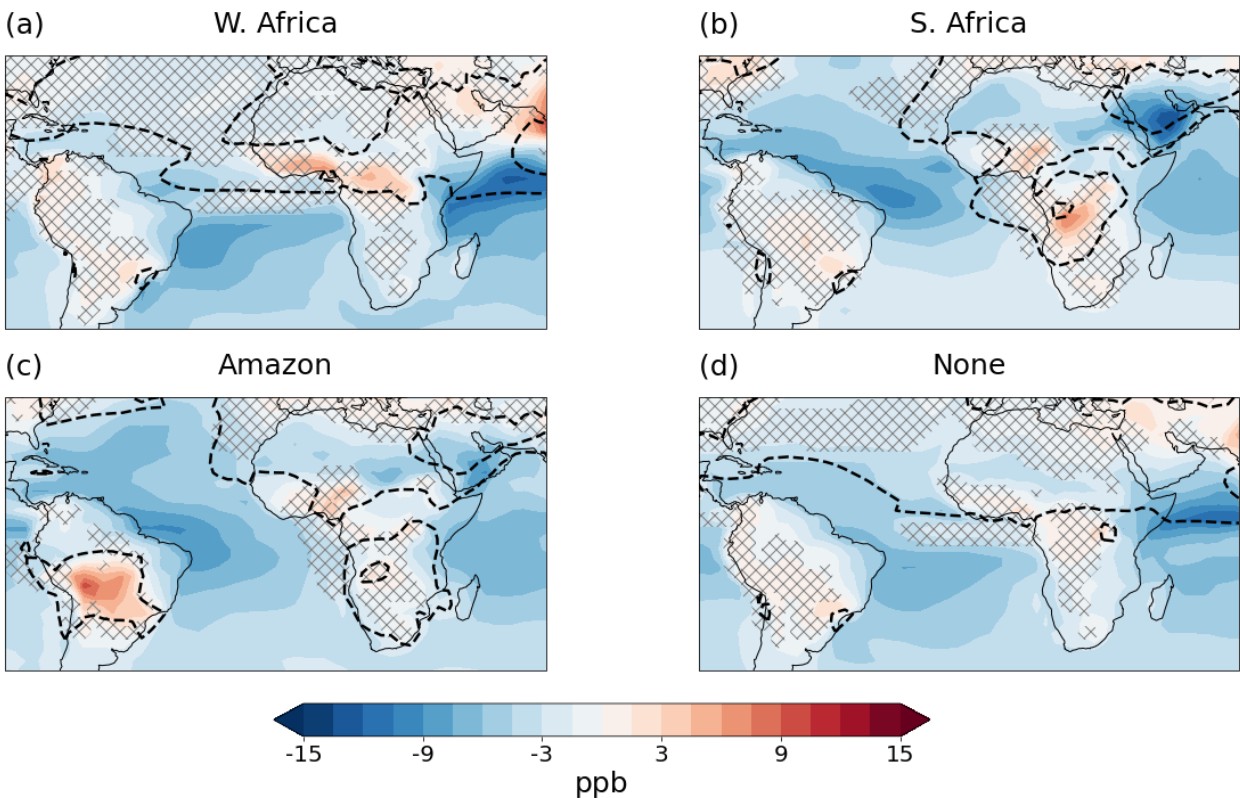

**Figure 4: The multimodel mean change in surface O₃ concentration due to climate change for the period 2090–2100 for (a) the Western African burning season (Dec–Feb), (b) the Southern African burning season (June, July), (c) the Southern Amazon burning season (Aug–Oct), and (d) the remaining months with limited burning (March–May, Nov). Grey hatching covers areas where models disagree on the sign of the change due to climate change. Black dotted lines outline areas where background O₃ is 40 ppb and 70 ppb.**


During December–February, the biomass burning area in Western Africa coincides with O₃ increases of 9 ppb (5 ppb for UKESM1, 7 ppb for MRI and 15 ppb for GISS; Fig. 4a) and similar O₃ penalties are seen for the Southern African biomass burning season during June–July (Fig. 4b). During the Amazon biomass burning season, there are even larger increases of up to 12 ppb in the Southern Amazon (Fig. 4c). In months without biomass burning, these areas have minor increases of 2 ppb

for UKESM1 and MRI and a decrease of 3 ppb for GISS.

GISS is the only model to show significant decreases in monthly mean surface O₃ concentration over land, which consistently occur in areas and seasons with low background O₃ (not shown). This includes biomass burning areas but in seasons without burning, which are followed by large increases in the biomass burning season. The result is that seasonal changes in surface

O₃ concentration due to climate change in GISS are much larger than UKESM1 and MRI, and models do not agree on the direction of change in remote areas, although the yearly average increase is similar between models (Fig. 3). This results in

uncertainty in the response to climate change from regions and seasons with low background $O_3$, but likely increases in areas and seasons with high background $O_3$ from anthropogenic NOx emissions (Fig. 4, black dashed lines).

Grid cells which include highly populated regions and megacities are often associated with an increase in $O_3$ concentration in all months, and an average ozone–climate penalty of 3 ppb in the yearly average. In particular, there is an ozone–climate penalty of 3 ppb that shows limited seasonal variation in grid cell containing the megacities in Nigeria (Lagos), Brazil (São Paulo, Rio de Janeiro) and Colombia (Bogotá, Medellín). This penalty is robust over Southeast Brazil in all seasons (Fig. 4).

### 3.4 Changes in chemical production and deposition of $O_3$ due to climate change


Attribution of changes in surface $O_3$ to changes in chemical production, chemical loss and dry deposition at the surface are shown in Fig. 5. The increase in $O_3$ production due to climate change is the largest out of these terms (Fig. 5c) and increases the most (over 0.25 Tg year$^{-1}$) in high-NOx areas where surface $O_3$ increases (Fig. 5a). Therefore, the increase in the rate of $O_3$ production is likely to be the main cause of the ozone–climate penalty in high-NOx areas (high-NOx defined as in Fig. 1).

Removal of $O_3$ by deposition and chemical destruction has a smaller effect on $O_3$ concentration since, to a degree, the two terms cancel each other out; in high-NOx areas, chemical loss increases by up to 0.1 Tg year$^{-1}$ and dry deposition decreases by up to 0.05 Tg year$^{-1}$. In remote regions, there is considerable variation between models as indicated by the higher standard deviation in these areas (Fig. 5, column 2). GISS predicts decreases in $O_3$ production over remote regions of up to 0.1 Tg year$^{-1}$ and increases of up to 0.25 Tg year$^{-1}$ over high-NOx regions, whereas MRI and UKESM1 predict increases in $O_3$ production

across all regions except Saharan Africa. MRI predicts the largest increases in $O_3$ chemical production in remote areas of 0.25 Tg year$^{-1}$ (not shown).

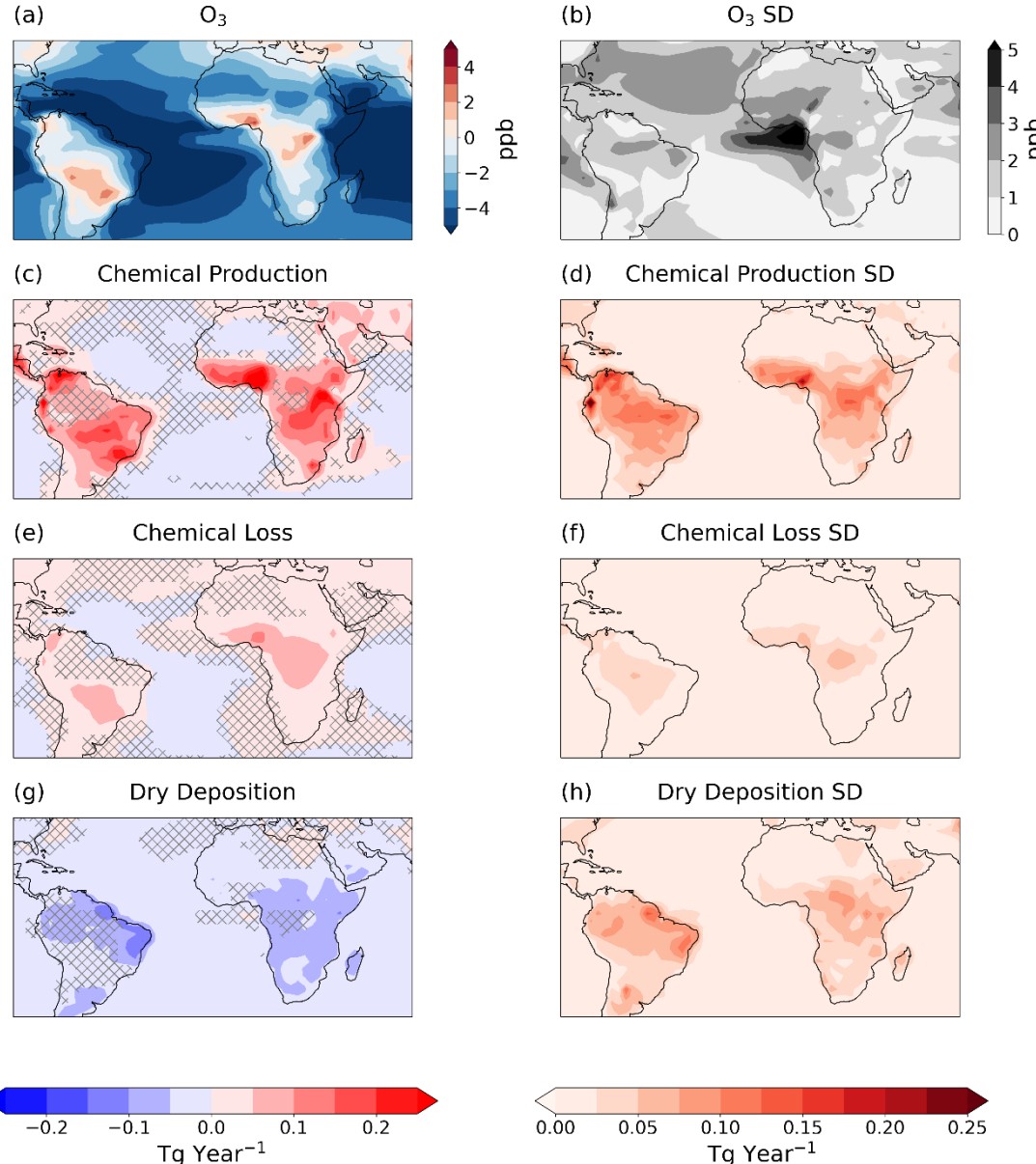

**Figure 5. The multi-model mean change in (a) surface O₃ concentration, (c) chemical production of O₃, (e) chemical destruction of O₃ and (g) dry deposition of O₃ due to climate change. Panels (c), (e) and (g) show the change in O₃ in Tg year⁻¹ and chemical terms have been summed over a 1 km height. The inter-model standard deviations are shown in panels (b), (d), (f), (h).**

Chemical loss and deposition changes become important in remote areas because these areas have the smallest increases in chemical production, but can have the largest changes in loss rate (Fig. S8) and deposition rate (Fig. S10). The rate of $O_3$ loss
is correlated with the change in isoprene concentration, which is typical of a low-NOx region due to reactions between isoprene and $O_3$ directly (Fig. S8). This leads to increases in the loss rate in most vegetated areas (Fig. S8). Conversely, the deposition rate decreases, presumably because the increased temperatures and lower relative humidity cause stomatal closure. Dry deposition varies between each model depending on stomatal response to temperature changes and boundary layer resistance changes (Fig. S10). In UKESM1, the increase in $CO_2$ also reduces stomatal conductance. UKESM1 shows a large decrease in
deposition rate over the central Amazon, whereas MRI shows very little change regionally.

In high-NOx areas, the increase in $O_3$ production is greater than the increase in loss leading to net chemical production of $O_3$. Evaluation of the sensitivity of $O_3$ chemical production rate to changes in isoprene emissions and NOx concentration due to climate change for each model is shown in Fig. 6 and described below.

Isoprene emission rate increases in both GISS and UKESM1 on average, as expected in a warmer climate (MRI does not have interactive isoprene emission) (Fig. 6, row 3). UKESM1 predicts a substantial decrease in isoprene emission rate in the Northern Amazon and fractionally in West Africa (Fig. 6a). Decreases in isoprene emission and stomatal conductance have previously been simulated in the same area due to $CO_2$ inhibition (Pacifico et al., 2012; Chadwick et al., 2017; Turnock et al.,
460    2020).

NOx decreases in most areas in UKESM1 including high-NOx areas, whereas GISS predicts increases of $2x10^{-11}$ to $6x10^{-11}$ mol $mol^{-1}$ in high-NOx areas only and MRI predicts more uniform increases of $4x10^{-11}$ mol $mol^{-1}$ in all areas (Fig. 6, row 2). The magnitude of the background NOx concentration in UKESM1 and the change due to climate change is also much larger
than the other models. The negative NOx concentration changes in UKESM1 and in some areas in GISS compared to MRI may be due to increased sequestration of NOx into isoprene nitrates. This possibility is supported by evidence of anticorrelation between NOx and isoprene in GISS and UKESM1 (Fig. 6). In GISS, the remote Amazon shows the largest isoprene increase and a decrease in NOx concentration. UKESM1 also shows an increase in NOx concentration downwind of the isoprene decrease in the Northern Amazon.

Despite differences in the magnitude and direction of the NOx and isoprene changes, the change in $O_3$ chemical production rate has similar spatial patterns in all models (Fig. 6, row 4). Exceptions occur in central Africa where GISS predicts a decrease in production rate, and in Nigeria where UKESM1 is the only model that does not predict a large increase in $O_3$ production. These areas of Africa also exhibit differences in surface $O_3$ concentration between models, discussed in Sect. 3.3.

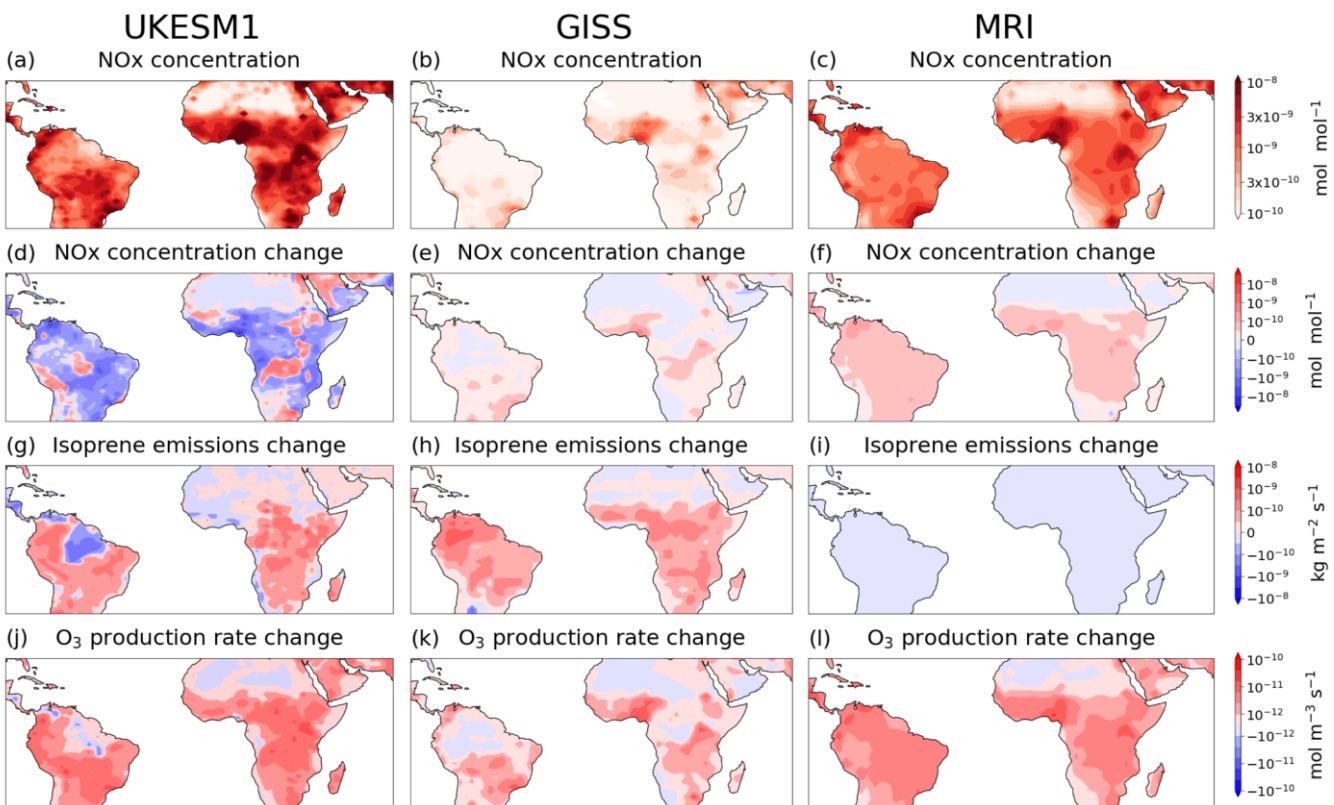

**Figure 6: (a), (b), (c)** Surface NOx concentrations in the absence of climate change and the average change due to climate change in **(d), (e), (f)** NOx concentration, **(g), (h), (i)** isoprene emission rate and **(j), (k), (l)** O$_3$ production rate for the period 2090–2100 for **(column 1) UKESM1, (column 2) GISS and (column 3) MRI.**


The O$_3$ production rate for GISS appears highly correlated with the change in NOx concentration in Fig. 6e, whereas NOx concentration decreases in many areas where O$_3$ production increases for UKESM1. Instead, isoprene may influence O$_3$ production in UKESM1. Areas with a decrease in isoprene emissions in UKESM1 also show a decrease or a smaller increase in O$_3$ production compared to other areas, suggesting isoprene is important for O$_3$ production in UKESM1 even in remote

regions such as the Northern Amazon.

To determine the strength of the relationship between O$_3$ production rate and changes in precursors NOx and isoprene, coefficients from a multiple linear regression are presented in Fig. 7. The monthly mean change in isoprene emission rate, NOx concentration and O$_3$ production rate for each grid cell are shown graphically with locations and months of high-NOx

(above the 95th percentile) marked with stars. All three climate models produce coefficients between 0.33 and 0.41 for the relationship between changes in NOx concentration and O$_3$ production rate (Fig. 7). However, the change in isoprene emissions using GISS and UKESM1 is a weaker predictor of O$_3$ production, even though increases in isoprene emission of over 100 % are predicted. All predictors are considered significant due to the large sample size, with r$^2$ values of 0.384, 0.732 0.590 for

UKESM1, GISS and MRI respectively (Table S3). The lower $r^2$ value for UKESM1 indicates that the changes in NOx
concentration and isoprene emissions explain less than half of the change in O$_3$ production rate. Additional analysis shows that
the O$_3$ production rate in UKESM1 is also related to the background NOx concentration (see S4: The relationship between
NOx and ozone production).

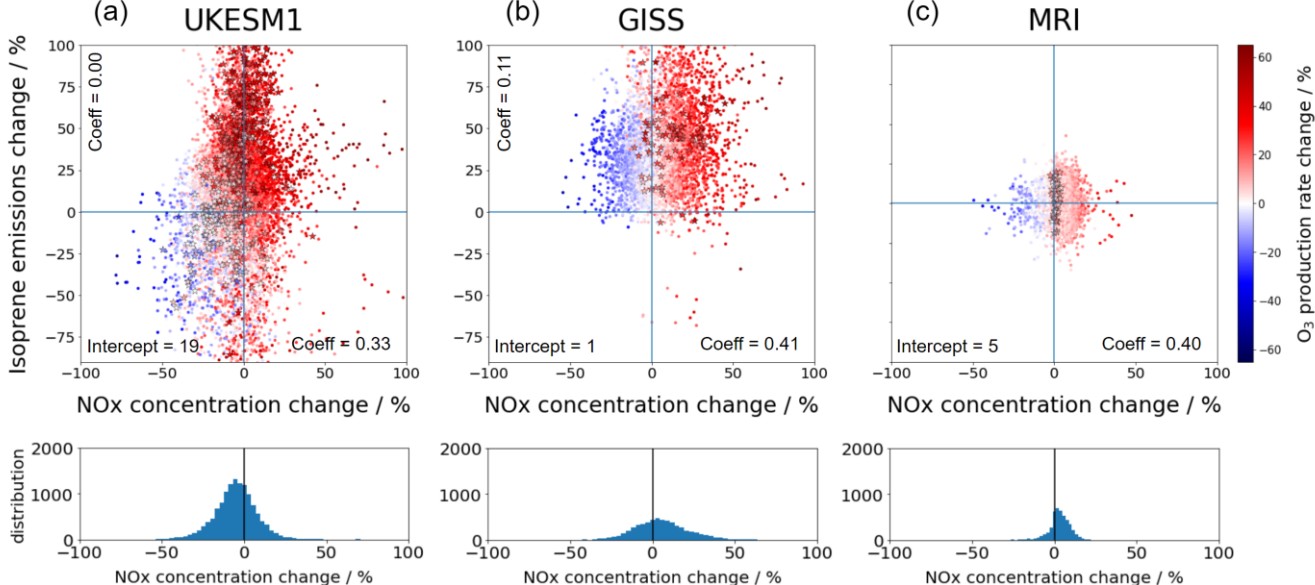

Figure 7: Scatter plots of the monthly mean percentage change in surface NOx concentration, isoprene emission rate and O$_3$
production rate for each grid cell and each month for (a) UKESM1, (b) GISS and (c) MRI for the region 30° S–30° N, excluding
Saharan Africa. Data for MRI has been randomly normally distributed along the y-axis. Grid cells and months where the
background NOx concentration is greater than the 95th percentile for the region shown in Fig. 3.1 are marked with stars. The
labelled intercept and coefficients refer to the results of a multiple linear regression ΔO$_3$ prod (%) ~ ΔNOx (%) + ΔIsoprene (%)
using the plotted data. The second row contains the number of data points in each NOx concentration change range. The data are
divided into 50 bins.

GISS simulates increases and decreases in NOx concentration of 50 %, compared to the smaller changes predicted by MRI,
which fall mostly in the range 0–20 % (Fig. 7c). GISS therefore predicts decreases in O$_3$ production over remote regions (Fig.
6b) and seasons, whereas MRI predicts consistent increases (Fig. 6c). Additionally, high-NOx areas simulated by GISS
experience an increase in O$_3$ production regardless of the NOx concentration change (Fig. 7b, stars). In high-NOx areas
simulated by UKESM1, the percentage change in NOx concentration is small so there is not enough information to identify
individual isoprene and NOx sensitivities, although areas with increased isoprene emission also show increases in O$_3$
production rate (Fig. 7a, stars).

The apparent relationship between O$_3$ and isoprene in UKESM1 (Fig. 6) does not show up using a linear model (Fig. 7). A
relationship may be hidden by variation in the isoprene–O$_3$ production sensitivity in different grid cells, or by correlations

between isoprene and NOx (discussed further in S4: The relationship between NOx and ozone production). However, as isoprene also contributes to $O_3$ loss, the effect of isoprene on net chemical production (production – loss) is reduced by the two terms cancelling each other out, so the change in net $O_3$ chemical production is more clearly related to the change in percentage NOx concentration (Fig. S7). In particular, the decrease in net $O_3$ production in the Northern Amazon (Fig. S7d) resembles the percentage change in NOx concentration (Fig. S7a) more than the percentage change in isoprene emissions (Fig. S7b).

The change in $O_3$ production rate will be further affected by meteorological changes, temperature in particular. This is the reason that $O_3$ production increases in UKESM1 and MRI even in the absence of changes in NOx and isoprene (the intercepts of the linear model are 19 % and 5 % respectively) and $O_3$ production increases in areas showing decreasing NOx concentrations in UKESM1. Since the temperature change varies seasonally and regionally, with dry seasons experiencing the largest increase in temperature, some of the changes in $O_3$ production in Fig. 7 may be driven by temperature rather than NOx or isoprene changes. If isoprene/NOx and $O_3$ production are both influenced by the underlying meteorology, the identified correlations may be due to meteorology rather than the chemical species changes. We verify that the monthly mean temperature change in each gridcell is not significantly correlated with percentage NOx change in any model, nor percentage isoprene change in UKESM1 (not shown). Therefore, NOx and isoprene changes are likely controlled by many processes in addition to temperature, including background chemistry and emissions for NOx, and vegetation type and cover for isoprene, as well as other meteorological variables. This indicates that the identified correlations between NOx and O3 production are unlikely to be the result of a spurious relationship driven by temperature, although it is still possible that the strength of the correlations may be inflated by confounding meteorological variables.

## 4. Discussion

When compared to in situ observations, the three climate models used in this study overestimate present-day surface $O_3$ in tropical regions by 14 ppb on average, including 11 ppb over the oceans. This is close to the global bias of 16 ppb calculated by Turnock et al. (2020) which included data from six climate models, including the three in this study. Therefore, the sources of error may not be unique to the tropics and subtropics. The major sources of variation between model and observations are related to differences in the area sampled and the heights of the stations relative to the lowest model grid cell (Pacifico et al., 2015).

In the tropics and subtropics, we expect in-canopy deposition and chemical processes to be the most important contributor to the positive bias because these processes create a steep $O_3$ gradient at the surface, whereas models aim to predict $O_3$ concentrations at 20+ m from the canopy top where these deposition processes are not included (Stroud et al., 2005; Gordon et al., 2014). Additionally, the volume of the model grid box is many times larger than the area sampled by measurement sites,

and also larger than the area of precursor emission sources such as fires. Therefore, the model inputs and predictions represent the average over a region that is not directly comparable with in situ measurements (Sinha et al., 2004). As a further validation, we also provide data from the TES satellite at 825 hPa, which records higher $O_3$ concentrations than the in situ sites since it measures at an altitude away from the canopy sink. Although this is much higher in altitude than the lowest grid box of any of the models, it should capture the above-canopy seasonal cycle at a resolution closer to the model grid resolution.


We find that the modelled surface $O_3$ bias compared to in situ observations is largest in biomass burning areas, although in South America the models capture the seasonal cycle well (Fig. 1). In situ sites, especially in the DR Congo (Fig. 1e), do not detect the large increases in $O_3$ predicted by models during biomass burning months although observed $O_3$ concentrations are also highest during biomass burning season (Adon et al., 2013). A high positive bias during the dry season has been found in 560 previous studies (e.g. Turnock et al., 2020) although our study has covered several regions that did not previously have available data. It is likely that effective removal within the canopy that is not included in models softens the observed seasonal cycle. In this region, the trends captured by satellites are closer to the model predictions, which increases confidence that models are correctly identifying $O_3$ enhancements above the canopy due to fires. However, future studies assessing the risks to human and ecosystem health should be aware of this limitation in current models.


The remainder of the study focuses on the change in surface $O_3$ due to climate change. Although model biases increase uncertainty in the change due to climate change, we quantify the difference between two simulations, which should remove systematic biases, and we note that the models capture seasonal and regional trends that are explained by either in situ measurements or satellite measurements (Fig. 1). This gives confidence in the trends in future $O_3$ change presented in this 570 study, but we highlight $O_3$ from biomass burning as an area for further study. The models have different chemistry schemes, land processes and temperature sensitivities which contribute to model variation (Stevenson et al., 2006; Wu et al., 2007; Archibald et al., 2020b). For this reason, we do not attempt to completely diagnose reasons for inter-model variation and instead the aim of this study is to identify robust predictions and areas of uncertainty for the change in $O_3$ due to climate change.


We find that while overall $O_3$ concentrations over the tropical land areas are reduced under the climate scenario examined here, climate change could lead to an ozone–climate penalty in areas which have a high background NOx concentration. These high-NOx areas already tend to have high $O_3$ concentrations in the absence of climate change (above 40 ppb), with climate change causing a further deterioration in air quality. Models predict that climate change will lead to seasonal mean increases 580 in surface $O_3$ concentration of up to 12 ppb in tropical and subtropical areas with high-NOx emissions (Fig. 4). The increase in surface $O_3$ in high-NOx areas is robust, with seasonal mean increases of up to 15 ppb for UKESM1, 18 ppb for GISS and 12 ppb for MRI. These areas are defined by NOx emission magnitudes above the 95th percentile for the region 40° S–40° N, which is dominated by anthropogenic contributions such as biomass burning or urban emissions. $O_3$ pollution in forested areas

has the potential to reduce forest productivity, decreasing the amount of carbon removed from the atmosphere and impairing forest resilience (e.g. Sitch et al., 2007; Grulke et al., 2020).

The ozone–climate penalty in high-NOx regions is primarily driven by an increase in $O_3$ chemical production, which is largest in areas of high-NOx (Figs 5, 6). This is in agreement with results from Doherty et al. (2013) and Archibald et al. (2020b) who showed that the rate of change of $O_3$ with temperature increases with NOx concentration. Firstly, the major $O_3$ forming reaction $NO + HO_2 / RO_2$, happens faster at higher temperatures and scaling up $O_3$ production in an area where $O_3$ production is already high will often lead to greater $O_3$ increases than an area with low $O_3$ precursor concentrations (Coates et al. 2016). Secondly, VOC and NOx concentrations can increase at higher temperatures. NOx concentration can increase due to increased PAN decomposition at higher temperatures (Doherty et al., 2013), changes in lightning frequency, or changes to atmospheric chemistry and VOCs increase largely as a result of increased isoprene emissions. All models likely exhibit an increase in reaction rates, however there were differences between UKESM1 and the other models in their sensitivity to changes in NOx concentration and isoprene emissions (Fig. 7).

Using a multiple linear regression, we find that changes in NOx concentration are strongly correlated with changes in $O_3$ production rate for GISS and MRI ($r^2$ = 0.732 and 0.590 respectively) (Fig. 7). For UKESM1, we find that linear regression using changes in NOx concentration and isoprene emission explains less than 50 % of the change in $O_3$ production rate. Including background NOx concentration in the linear regression improves the $r^2$ value and suggests that the rate of $O_3$ production increases in proportion to the background NOx concentration (Fig. S5). UKESM1 has previously been identified as being among the least responsive to changes in precursor concentrations out of the CMIP6 models (Turnock et al., 2020), and indeed chemical production increases in many areas despite decreases in NOx, so the increase in chemical $O_3$ production is more likely to be dominated by an increase in rate of reaction not changes in precursor concentration. The linear regression uses monthly means, so modelled $O_3$ increases during burning seasons (dry seasons) are likely compounded by the fact that these seasons often show the greatest temperature increase due to climate change (Fig. S3). Therefore, some of the identified correlations may be due to meteorology changes rather than chemical changes.

As changes in NOx concentrations are shown to be important for changes in $O_3$ production in GISS and MRI, we now discuss the inter-model differences in NOx concentration changes in further detail. GISS and MRI agree that NOx concentrations will increase in high-NOx regions, but disagree on the direction of change in remote regions. UKESM1 predicts a decrease in NOx concentrations in many areas. GISS predicts a decrease in NOx of $2x10^{-11}$ mol mol$^{-1}$ in remote areas (up to 50 %) while MRI predicts an increase of $2x10^{-11}$ to $6x10^{-11}$ mol mol$^{-1}$ in remote regions. This is likely causing the difference in $O_3$ production between GISS and MRI in remote regions. To reduce uncertainty in predictions for $O_3$ concentration changes due to climate change, further work to constrain future NOx concentration changes is needed.

The NOx concentration change depends on the balance of NOx production and loss terms. A large contributor to increases in NOx concentrations in all models is an increased decomposition of PAN into NOx, which will be largest in source regions. Lightning NOx is also a NOx source, but its influence on surface NOx remains unclear. Although lightning NOx increases in all models during the wet season, the largest surface NOx and $O_3$ increases occur in the dry season, so the ozone–climate penalty is unlikely to be driven by lightning NOx changes. Nevertheless, the large increase in lightning NOx in MRI may have a role in the increase in surface NOx concentration in MRI, which is larger than the other models, and lightning NOx decreases in the Northern Amazon during the dry season (Fig. S4a) may contribute to the decrease in NOx and $O_3$ production in this region in UKESM1 (Fig. S7). A large contributor to the loss term will be reaction with isoprene derivatives, thus increased formation of isoprene nitrates. In both UKESM1 and GISS, isoprene and NOx are anti-correlated in some areas, suggesting isoprene emissions changes have a notable effect on NOx concentrations. For example, GISS predicts large isoprene increases in the remote Amazon, where the major NOx decreases occur, and UKESM1 shows a small area of increased NOx concentrations downwind of the Northern Amazon, where isoprene decreases. As MRI prescribes isoprene as a climatology, there is will be no significant change to NOx loss via organic nitrate formation and this is likely a reason why NOx increases over most areas of land. The reasons why NOx loss dominates in UKESM1 whereas GISS shows a net NOx increase requires further understanding of individual model details such as isoprene nitrate yield and NOx recycling frequency. The sensitivity of NOx concentration to changes in lightning, PAN and isoprene in each model is beyond the scope of this study, but would be useful to explore in further work. Further studies could also explore some temperature-sensitive sources of NOx that were not included in the simulations such as soil NOx emissions and changes in wildfire frequency.

NOx is not the only driver of changes in $O_3$ production, and changes in temperature-dependent emissions of isoprene also influence the rate. This may be especially important in UKESM1, which predicts the highest concentration of background NOx, because high-NOx areas may experience a different chemical regime in which VOCs also increase $O_3$ production (e.g. Liu et al., 2013). For example, UKESM1 shows increases in $O_3$ production rate as isoprene increases in high-NOx areas (Fig. 7, stars). However, calculating the net effect of isoprene on surface $O_3$ concentrations is complex; increasing isoprene emissions can increase both the rate of $O_3$ production and the rate of $O_3$ loss and, as described above, can decrease concentrations of NOx and OH. Examining the percentage change in net $O_3$ chemical production in UKESM1 (Fig. S7) suggests the net effect of changing isoprene emissions on $O_3$ chemistry cancel out, and that the percentage change in net $O_3$ production is more closely related to percentage changes in NOx concentration.

The increase in chemical loss is strongly correlated with isoprene concentration change in UKESM1 and GISS (Fig. S8). Therefore, the different isoprene schemes used by each model contributes to uncertainty in the loss rate over the continents. In particular, MRI used climatological isoprene resulting in no significant change in the loss rate (Fig. S8c) and UKESM1 includes $CO_2$ inhibition, which decreases the isoprene emission rate and loss rate in in Northern Amazonia (Fig. S8a). Overall,

this means that GISS has a higher loss rate, especially in low-NOx, high isoprene regions such as the remote Amazon, which may partly account for the larger decreases in $O_3$ in remote regions using this model (Fig. 3).

The global study by Zanis et al. (2022) employs two additional climate models and also finds climate benefits and uncertainties in surface $O_3$ concentrations in the same remote regions. We excluded these two models from our own study as data for the sensitivity study (NOx concentration and isoprene emission rate) were not available at the time of writing. Zanis et al. (2022) highlight the different isoprene emission schemes as a reason for model variation however our analysis (Fig. 7) finds NOx to be the most important precursor. The fact that the models contain positive correlations between the change in NOx and $O_3$ production, and between the change in isoprene and $O_3$ loss, indicates the tropics and subtropics exhibit NOx-limited behaviour, although isoprene may be important in high-NOx areas. We agree that isoprene is highly relevant for the change in loss rate and for indirect effects on $O_3$ through changes in related atmospheric chemistry such as OH and NOx concentrations.

The decrease in deposition rate is controlled by UKESM1 and GISS (MRI showed very little change), but there was spatial variation in the magnitude of the change. This could be due to changes in meteorology between models (such as temperature and precipitation), as well as model differences (UKESM1 includes $CO_2$ inhibition). Feedbacks such as $O_3$ damage to vegetation were not considered in any model (Pacifico et al., 2015) but may be a useful addition to future simulations.

Models tend to predict a decrease in surface $O_3$ over regions strongly affected by ocean air such as North Brazil. This is due to robust decreases in $O_3$ over the oceans from increases in atmospheric water vapour. Over land, increases in water vapour and OH influence the concentrations and lifetimes of many species.

We finally note that Nigeria experiences substantial increases in $O_3$ production according to GISS and MRI, whilst a slight decrease is predicted using UKESM1. This is an important geographical area for future research since poor air quality could affect large numbers of people living in West African cities. In this area, the choice of emissions scenario is also important for determining the $O_3$ response to climate change because NOx emissions in Nigeria increase rapidly in the SSP3-7.0 scenario compared to the present-day due to predicted urbanisation. Therefore, future studies should explore alternative emissions pathways to better inform policy.

## 5.  Conclusion

Using a multimodel mean of data from three Earth system models, we identify that by 2100, there will be an ozone–climate penalty in high-NOx areas, such as major cities and biomass burning areas (Fig. 4). This is not due to increased fire emissions, but due to the increasing temperature, which speeds up the recycling of NO into NOx and increases decomposition of PAN into NOx in source regions. It shows that the ozone–climate penalty is greatest in areas already experiencing high $O_3$, putting

forests in these areas at greater risk of $O_3$ damage and urban populations at increasing threat of health problems. This study adds to findings from the World Health Organisation World Air Quality Report (2021) that air pollution is an increasing issue across the tropics, and that there is a need for greater monitoring of air pollution across Africa and South America.

The Earth system models display NOx-limited behaviour, including that higher NOx concentrations lead to increased $O_3$ chemical production and therefore increased surface $O_3$ concentration (Figs 5–7). As the background concentrations of NOx are largely anthropogenic, this suggests that without reduction in emissions, forested areas in urban and fire-prone locations are more at risk from increases in surface $O_3$ due to climate change than remote forests. As $O_3$ damage can reduce plant productivity, this has implications for the success of secondary forests and other human-modified forests which are mostly located in agricultural areas, deforestation frontiers and forest edges (Heinrich et al., 2021), and may reduce their carbon sequestration potential (Sitch et al., 2007).

In remote regions, differences in the direction of $O_3$ concentration change between models creates uncertainty as to whether remote locations are at greater risk of $O_3$ damage in a warmer climate, although ocean influenced areas display robust climate benefits (Fig. 4). Further work is needed to constrain the climate response of isoprene emissions and the temperature sensitivity of NOx and $O_3$ chemistry.

**Data availability statement**

All CMIP6 model data used in the present study can be obtained from https://esgf-node.llnl.gov/search/cmip6/.

All TOAR I data used in the study can be obtained from https://join.fz-juelich.de/services/rest/surfacedata/.

All INDAAF data used in this study can be obtained from http://www.indaaf.obs-mip.fr

**Author contributions**

FB wrote the paper and led the data analysis with contributions from all authors. SS and GAF contributed to the interpretation of the data. MB contributed to the statistical analysis. IDS, HV, PB, MB and CGL contributed in situ data for model evaluation. SEB and KT contributed in the GISS-E2-1-G simulations; MD and NO contributed in the MRI-ESM2-0 simulations; JK and FMO contributed in the UKESM1-0-LL simulations.

**Competing interests**

The authors declare that they have no conflict of interest.

## Acknowledgements

FB was funded by the NERC GW4+ DTP (award no. NE/S007504/1) and the Met Office on a CASE studentship. LMM acknowledges funding from the UK Natural Environment Research Council funding (UK Earth System Modelling Project, UKESM, Grant no. NE/N017951/1). SS, LMM and AWC were supported by NERC funding (Grant no. NE/R001812/1). MD and NO were supported by the Japan Society for the Promotion of Science KAKENHI (Grant no. JP18H03363, JP18H05292, JP19K12312, JP20K04070 and JP21H03582), the Environment Research and Technology Development Fund (JPMEERF20202003 and JPMEERF20205001) of the Environmental Restoration and Conservation Agency of Japan, the Arctic Challenge for Sustainability II (ArCS II), Program Grant Number JPMXD1420318865, and a grant for the Global Environmental Research Coordination System from the Ministry of the Environment, Japan (MLIT1753). JK was financially supported by NERC through NCAS (Grant no. R8/H12/83/003), and the Met Office CSSP-China programme funded POZsUM project. JH, FMO'C and GAF were supported by the Met Office Hadley Centre Climate Programme funded by BEIS. FMO'C and GAF were supported by the EU Horizon 2020 Research Programme CRESCENDO project (Grant no. 641816). Resources supporting the GISS simulations were provided by the NASA High-End Computing (HEC) Program through the NASA Center for Climate Simulation (NCCS) at the Goddard Space Flight Center. GISS authors, SEB and KT acknowledge funding from the NASA Modeling and Analysis program. IV was funded by the Fonds Wetenschappelijk Onderzoek Flanders (FWO; grant no. G018319N). The INDAAF project is funded by the INSU/CNRS/IRD and is part of the ACTRIS-FR Research Infrastructure. The authors are grateful to the field and chemistry analytical technicians who operate the African sites and help to provide reliable data.

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
