# Peer review of "The ozone-climate penalty over South America and Africa by 2100"

_EGUsphere, 2022_

## Author Comment (AC1)

Author response:

We thank the editor and reviewers for their constructive and detailed comments, which have helped us to improve our manuscript. All comments have been addressed. The reviewer comment is shown in black, our reply is shown in blue and extracts from the revised manuscript text are shown in 'commas', with altered text in red and deleted text with .

Please note that a large section of the discussion has been rearranged to incorporate a discussion of NOx changes (so shows up in red even though it has not been completely rewritten).

REVIEWER 1:

This study nicely illustrates the impact of NOx levels in determining whether regions within South America and Africa are likely to incur an ozone-climate penalty. This is useful work and should be suitable for publication after accounting for the suggestions below. There is a robust correlation between isoprene emission changes and ozone production. A signal of correlation of increased ozone production with increased NOx change is found, and in one model there is a correlation with absolute NOx levels.

We thank the reviewers for their interest in the paper. We appreciate comments suggesting a greater discussion as to the cause of NOx changes and the cause of the ozone-climate penalty and have developed the discussion to elevate the paper in this way. The changes are described below and we hope that this is now covered in sufficient detail.

Further to the changes shown here, the discussion and Sect. 3.4 have been rearranged to incorporate the points made by reviewers and improve flow.

Reasons for the NOx changes are not explored, these could be due to changes in wet/dry deposition, changes in emission (lightning is mentioned, but not explored) or changes in organic nitrate and PAN formation. The latter are mentioned in the introduction, but not followed up in the discussion. In figure 6 (particularly UKESM1, but also GISS) the NOx and Isoprene emission changes seem anticorrelated over S. America, and possibly also over Africa. This would support the increased sequestration of NOx in organic nitrates and PANs.

These are all excellent points. We have included a greater discussion of lightning NOx, PAN and organic nitrate formation (including fig 4 in the supplementary).

[Figure]

**Figure S4: The change in lightning NOx column for the period 2090–2100 for (a, b, c) Dec–Feb, (d, e, f) Jun–Aug for (column 1) UKESM1, (column 2) GISS and (column 3) MRI.**

However, we are unable to fully identify the reason for the NOx changes in each model. Whilst the paper aims to identify (A) robust changes in climate-driven ozone concentration, (B) partition the change into chemical and deposition changes, (C) suggest the role of NOx and VOCs in the chemical changes and (D) highlight the large variation in model prediction of NOx and VOCs, it is not intended to fully diagnose the cause of NOx and VOC differences between models. Further sensitivity simulations to diagnose the reason for the difference in NOx concentrations between models goes beyond the scope of this study but may be the focus of future work.

Addition to Sect. 3.2 (from line 330):

'NOx emissions, including biomass burning emissions, are prescribed based on the SSP3-7.0 scenario but lightning NOx and soil NOx differs between the models based on the chosen parameterisation of individual models. Compared to the present-day, NOx emissions in biomass burning areas decrease in Africa to follow projected trends, but do not change in South America. NOx emissions increase in cities and Nigeria especially has major growth in urban areas. Compared to the scenario without climate change, total lightning NOx emissions increase in all models, and the increases occur during the wet season (Fig. S4). MRI predicts much larger increases than GISS and UKESM1, and UKESM1 shows a decrease in lightning NOx over the Amazon basin in DJF (Fig. S4a) although the net effect over all seasons is positive. PAN decreases in all models (-94 ppt, -61 ppt, -30 ppt for UKESM1, GISS and MRI respectively) due to increased thermal decomposition. In GISS and UKESM1, the increase in isoprene emissions can increase removal ofes NOx via formation of isoprene nitrates. Soil NOx does not change in response to climate change in any model. The formation of isoprene nitrates may be more effective at removing NOx in UKESM1 compared to GISS, driving an overall decrease in NOx concentrations, Additionally, lightning NOx emissions can change

 '

Additions to Sect. 3.4 (from line 417):

' NOx decreases in most areas in UKESM1 including high-NOx areas, whereas GISS predicts increases of $2\times10^{-11}$ to $6\times10^{-11}$ mol mol$^{-1}$ in high-NOx areas only and MRI predicts more uniform increases of $4\times10^{-11}$ mol mol$^{-1}$ in all areas (Fig. 6, row 2). The magnitude of the background NOx concentration in UKESM1 and the change due to climate change is also much larger than the other models. The negative NOx concentration changes in UKESM1 and in some areas in GISS compared to MRI may be due to increased sequestration of NOx into isoprene nitrates. This possibility is supported by evidence of anticorrelation between NOx and isoprene in GISS and UKESM1 (Fig. 6). In GISS, the remote Amazon shows the largest isoprene increase and a decrease in NOx concentration. UKESM1 also shows an increase in NOx concentration downwind of the isoprene decrease in the Northern Amazon. '

These points are brought together in the discussion to give an overall view of NOx changes in each model.

' The NOx concentration change depends on the balance of NOx production and loss terms. A large contributor to increases in NOx concentrations in all models is an increased decomposition of PAN into NOx, which will be largest in source regions. Lightning NOx is also a NOx source, but its influence on surface NOx remains unclear. Although lightning NOx increases in all models during the wet season, the largest surface NOx and $O_3$ increases occur in the dry season, so the ozone–climate penalty is unlikely to be driven by lightning NOx changes. Nevertheless, the large increase in lightning NOx in MRI may have a role in the increase in surface NOx concentration in MRI, which is larger than the other models, and lightning NOx decreases in the Northern Amazon during the dry season (Fig. S4a) may contribute to the decrease in NOx and $O_3$ production in this region in UKESM1 (Fig. S7). A large contributor to the loss term will be reaction with isoprene derivatives, thus increased formation of isoprene nitrates. In both UKESM1 and GISS, isoprene and NOx are anti-correlated in some areas, suggesting isoprene emissions changes have a notable effect on NOx concentrations. For example, GISS predicts large isoprene increases in the remote Amazon, where the major NOx decreases occur, and UKESM1 shows a small area of increased NOx concentrations downwind of the Northern Amazon, where isoprene decreases. As MRI prescribes isoprene as a climatology, there is will be no significant change to NOx loss via organic nitrate formation and this is likely a reason why NOx increases over most areas of land. The reasons why NOx loss dominates in UKESM1 whereas GISS shows a net NOx increase requires further understanding of individual model details such as isoprene nitrate yield and NOx recycling frequency. The sensitivity of NOx concentration to changes in lightning, PAN and isoprene in each model would be useful to explore in further work. Further studies could also explore some temperature-sensitive sources of NOx that were not included in the simulations such as soil NOx emissions and changes in wildfire frequency. '

It would be useful to look at P-L to determine whether the isoprene emissions have a net positive or negative impact on ozone. The introduction implies that the sign of the net effect depends on the NOx background whereas in figure 6 for UKESM1 the

areas of increased isoprene emission all seem to have increase ozone production. Would the balance become more obvious when looking at P-L?

In relation to UKESM1, the change in net O3 production seems driven by NOx rather than isoprene emission. There is no clear relationship between isoprene emissions change and net O3 production. P-L in remote areas with isoprene increases is smaller than P because of an increase in loss rate L. In the Northern Amazon, where isoprene decreases, the lower rate of ozone production, P, over the Northern Amazon does not appear in the P-L figure, since loss rate also decreases. This is now shown in the supplementary (Fig. S7d). Some altered text and figure S7 is shown in this document in response to the comment beginning 'Figure 7: The correlation patterns here don't look to be quite the same as judging the correlations by eye from figure 6.'

As GISS shows very little sensitivity to isoprene emission we feel it is not necessary to include P-L for GISS or MRI in the supplementary.

Using monthly values in figure 7 might lead to spurious correlations. There will be strongly seasonal variations in isoprene and NOx, and also strong seasonal variations in meteorology (wet vs dry). Some of the correlations in figure 7 might be due to the meteorology -i.e. isoprene emission changes might be stronger in the dry season where the meteorological impacts on ozone might also be more positive if dry gets drier.

This is true; the O3 production term is simply the rate of the reaction NO + VOC/HO2. Therefore, ozone production can be changed by changes in NOx, VOC or the rate constant. As the reviewer states, meteorological changes will change k, NOx and VOC concentrations. This figure ignores changes in k, which is an oversimplification and may lead to spurious correlations if k is strongly correlated with isoprene or NOx.

One possible solution is to use yearly means, which we tested and the results were very similar (producing the same trends in correlation coefficients and $r^2$ values) however this still has the issue that there are regional variations in mean annual meteorology. When the variation in temperature change is separated into regional variation and seasonal variation, both latitude and season contribute equally to the temperature variation. This means that using the mean annual meteorology will not remove variations in isoprene and NOx driven by meteorology.

Another option is to check whether there are any strong relationships between temperature change and NOx/isoprene/O₃ production, assuming temperature change is the most important meteorological driver of k. The only relationship we identify is between temperature and isoprene in the GISS model, however the correlation coefficient between isoprene and ozone production is very small so this does not change our conclusions (i.e. our conclusion is that NOx is more important than isoprene for ozone production in GISS). One of the ways we have reduced the impact of seasonal meteorology on the correlations is by looking at percentage changes. This takes into account the background state to some extent so the change in NOx/isoprene is no longer necessarily related to the seasonal cycle.

That said, increasing temperature definitely plays a role in ozone production, since he intercept of the linear model in UKESM1 and MRI is positive, i.e. rate increases even in the absence of changes in isoprene and NOx. Therefore, by excluding meteorological changes

from the analysis we are missing some information. However, the aim of Fig. 7 was to determine the relative importance of NOx compared to isoprene and we feel this has been achieved without needing to modify the figure.

To make the caveat clearer, we have added further details on the role of meteorology in sect. 3.4:

'The change in $O_3$ production rate will be further affected by meteorological changes, temperature in particular. This is the reason that $O_3$ production increases in UKESM1 and MRI even in the absence of changes in NOx and isoprene (the intercepts of the linear model are 19 % and 5 % respectively) and $O_3$ production increases in areas showing decreasing NOx concentrations in UKESM1. Since the temperature change varies seasonally and regionally, with dry seasons experiencing the largest increase in temperature, some of the changes in $O_3$ production in Fig. 7 may be driven by temperature rather than NOx or isoprene changes. If isoprene/NOx and $O_3$ production are both influenced by the underlying meteorology, the identified correlations may be due to meteorology rather than the chemical species changes. We verify that percentage NOx change is not related to temperature in any of the models and that percentage isoprene change is not related to temperature in UKESM1 (not shown), which indicates the identified correlations are related to chemical species changes not meteorological variation, although this cannot be entirely ruled out.'

Line 58-60: This sentence starts with the effect of ozone on climate, but the references are all to the effect of climate on ozone. Ozone doesn't lead to a positive forcing through increases in anthropogenic precursors, it is through its absorption and emission of longwave radiation.

We thank the reviewer for spotting this error. We agree, and have changed the sentence accordingly

' Additionally, $O_3$ is a near-term climate forcer with impacts on the radiative balance leading to a positive radiative forcing of climate through absorption of longwave radiation (Myhre et al., 2017)'

Line 82: "Biogenic isoprene is the major O3-forming NMVOC …" This seems to imply that more ozone is formed from isoprene than from other NMVOCs. Is this true? Is this globally or just over forests? Does this mean gross formation i.e. ISOPOO +NO dominates the sum of RO2+NO, and ignoring the sinks. Elsewhere it is not clear even whether isoprene is a net producer of ozone.

This has been changed as the intention was to highlight isoprene as an important VOC in the tropics rather than to argue its net effect on ozone production.

'The sensitivity of $O_3$ production to NOx depends on the relative concentrations of NOx and VOCs. Isoprene is the most abundant biogenic VOC in remote Africa and South America and must be oxidised in the atmosphere before it can form $O_3$ (Liu et al., 2016).'

Line 86: It is mostly OH+NO2->HNO3 that causes NOx-saturation.

The paragraph now describes the basics of NOx-limited and NOx-saturated regimes.

' Regions are defined as NOx-limited when increasing  VOCs or OH acts to reduce $O_3$ concentrations through oxidation and formation of  organic peroxides  (Pacifico et al., 2012). In this NOx-limited case, increasing NOx will lead to greater $O_3$ formation. Conversely, in  VOC-limited regions with sufficient NOx present, increasing NOx concentrations may reduce $O_3$ concentrations by removal of the key $O_3$-forming radicals OH (reaction: $OH + NO_2 \rightarrow HNO_3$).  '

Line 115: This could be described better by explicitly saying that O1D+H2O is the major ozone loss.

We agree with the reviewer.

'Studies agree that over the ocean, average surface $O_3$ concentrations will decrease under the influence of climate change  (Zeng et al., 2008; Doherty et al., 2013; Zanis et al., 2022). The warmer air can hold more water vapour, a major species contributing to $O_3$ loss (reaction: $O(1D) + H_2O \rightarrow 2OH$ leading to $O_3$ loss via $OH_2 + O_3 \rightarrow OH + 2O_2$).  '

Section 2.1: Are the Price and Rind parameterisations in UKESM1 and MRI the same? Thornhill et al. 2021 found different climate responses from schemes that were all supposed to be Price and Rind. Soil NOx is mentioned, do these models include climate-dependent soil NOx emissions?

We now show lightning NOx in the supplementary. The large differences in lightning NOx response between UKESM1 and MRI suggests there are differences between the schemes, but details are probably not needed since this study does not ascertain the importance of lightning in the ozone production response.

Soil NOx is calculated individually for each model but does not respond to changes in climate in any model.

Text added to line 330:

'NOx emissions, including biomass burning emissions, are prescribed based on the SSP3-7.0 scenario but lightning NOx and soil NOx differ between the models based on the chosen parameterisation of individual models. Compared to the present-day, NOx emissions in biomass burning areas decrease in Africa to follow projected trends, but do not change in South America. NOx emissions increase in cities and Nigeria especially has major growth in urban areas. Compared to the scenario without climate change, total lightning NOx emissions increase in all models, and the locations of the increases vary in latitude to follow the wet season (Fig. S4). MRI predicts much larger increases than GISS and UKESM1, and UKESM1 shows a decrease in lightning NOx over the Amazon basin in DJF (Fig. S4a) although the net effect over all seasons is positive. PAN decreases in all models (-94 ppt, -61 ppt, -30 ppt for UKESM1, GISS and MRI respectively)  due to increased thermal decomposition. In GISS and UKESM1, the increase in isoprene emissions can increase removal of NOx via formation of isoprene nitrates. Soil NOx does not change in response

to climate change in any model.  '

Line 234: It might be clearer to state that fixing CO2 in pd means that the difference compared to the baseline includes the effect of CO2 inhibition.

Definitely. This has been changed as follows:

'In UKESM1, $CO_2$ is also fixed to present-day concentrations in ssp370pdSST so that the effect of climate change includes the effect of $CO_2$ inhibition. '

Line 241: Are the model levels really "above the canopy" or are they above the orography?

Changed to orography.

Line 386: Might be clearer to say that Emissions from cities create positive sensitivities to climate change in all months.

Adapted also in response to reviewer 2 who pointed out that the resolution of the model does not represent individual cities.

'  Grid cells which include highly populated regions and megacities  often associated with an increase in $O_3$ concentration in all months, and an average ozone–climate penalty of 3 ppb in the yearly average. In particular, there is an ozone–climate penalty of 3 ppb that shows limited seasonal variation in grid cell containing the megacities in Nigeria (Lagos), Brazil (São Paulo, Rio de Janeiro) and Colombia (Bogotá, Medellín).  This penalty is robust over Southeast Brazil in all seasons (Fig. 4).'

Line 414: The plot of loss frequency is very useful.  It would also be useful to show dry deposition as a velocity in the supplement i.e. dividing by ozone concentration in kg/m3.

Done (Fig. S10 is shown on page 18)

Line 414: In UKESM1 CO2 will also affect the stomatal closure, I don't know about the other models.

Indeed this is true for UKESM1 only. Further comments have been added to this effect, also in response to reviewer 2.

'Conversely, the deposition rate decreases, presumably because the increased temperatures and lower relative humidity cause stomatal closure. Dry deposition varies between each model depending on stomatal response to temperature changes and boundary layer resistance

changes (Fig. S10). In UKESM1, the increase in $CO_2$ also reduces stomatal conductance. UKESM1 shows a large decrease in deposition rate over the central Amazon, whereas MRI shows very little change regionally. '

Figure 7: The correlation patterns here don't look to be quite the same as judging the correlations by eye from figure 6. This might be because this uses month as another variable. Is the isoprene coefficient really 0.00 for UKESM1? It looks as if higher isoprene leads to higher ozone production, both in figure 7 and in 6.

The isoprene coefficient is 0. Likely, the sensitivity to isoprene varies regionally so the correlation coefficient varies by grid cell and leads to an overall average of zero. As the reviewer notes, ozone production rate looks related to isoprene emissions in Fig. 6 especially in areas where isoprene decreases, even though the model does not pick up on this. However, these areas also have the largest percentage decrease in NOx, which cannot be seen in Fig. 6 because it shows absolute changes (but Fig. S7 has been added to the supplementary to show this, see below) so it's very hard to tell exactly what is going on 'by eye'. No doubt isoprene plays a role in these areas and this is why Fig. 6 is needed as well as Fig. 7, and we have amended the text to reflect this more clearly.

Unfortunately, adding month as a random variable is unlikely to improve correlations. The wet and dry seasons depend on latitude so occur during different months depending on the grid cell. Therefore, March in one place has very little in common with March elsewhere so seasonally-driven variation will not be removed.

We have added further discussion on isoprene in UKESM1 (starting from line 437).

'The $O_3$ production rate for GISS appears highly correlated with the change in NOx concentration in Fig. 6e, whereas NOx concentration decreases in many areas where $O_3$ production increases for UKESM1. Instead, isoprene may influence $O_3$ production in UKESM1. Areas with a decrease in isoprene emissions in UKESM1 also show a decrease or a smaller increase in $O_3$ production compared to other areas, suggesting isoprene is important for $O_3$ production in UKESM1 even in remote regions such as the Northern Amazon.

…

The apparent relationship between $O_3$ and isoprene in UKESM1 (Fig. 6) does not show up using a linear model (Fig. 7). A relationship may be hidden by variation in the isoprene - $O_3$ production sensitivity in different grid cells, or by correlations between isoprene and NOx (discussed further in S4: the relationship between NOx and ozone production). However, as isoprene also contributes to $O_3$ loss, the effect of isoprene on net chemical production (production – loss) is reduced by the two terms cancelling each other out, so the change in net $O_3$ chemical production is more clearly related to the change in percentage NOx concentration (Fig. S7). In particular, the decrease in net $O_3$ production in the Northern Amazon (Fig. S7d) resembles the percentage change in NOx concentration (Fig. S7a) more than the percentage change in isoprene emissions (Fig. S7b).'

The text and figure from the supplementary:

'The low correlation between UKESM1 ozone production rate and isoprene emissions is surprising given the apparent spatial correlations in Fig. 6 (column 1). In particular, the lower production rate in the Northern Amazon appears related to the decrease in isoprene emissions in the same area. If the change in NOx, isoprene and $O_3$ production is shown spatially as a percentage change as in Fig. S7, it becomes clear that the Northern Amazon also has a percentage large reduction in NOx concentration (Fig. S7a), and that the areas of decreased ozone production are associated with these large NOx concentration decreases. These NOx decreases may be due to lightning changes, since this area experiences decreases in column lightning NOx in several months (Fig. S4). Even so, changes in $O_3$ production may still be related to isoprene emissions changes in UKESM1 over the Amazon. The linear model may not identify the relationship if NOx and isoprene are also correlated or if the strength or direction of the correlation between isoprene emissions and $O_3$ production is different in different areas. As it is already known that the concentration of background NOx can influence the role of isoprene in $O_3$ production (i.e. NOx-limited vs VOC-limited regimes), it is likely that isoprene emissions are important and the linear model does not give the details. Rather than investigate this further, we show in Fig. S7d that in terms of net $O_3$ production (the focus of the study), the effect of isoprene emissions changes cancel out. As isoprene can increase the $O_3$ loss rate and production rate, the net change in $O_3$ chemical production has more in common with the NOx concentration change.

**UKESM1**

(a)    % NOx concentration change

(b)    % Isoprene emissions change

(c)    % O$_3$ production rate change

(d)  % net O$_3$ production rate change

[Figure]

−0.9    −0.6    −0.3    0.0    0.3    0.6    0.9
%

**Figure S7: The annual mean percentage change due to climate change in (a) NOx concentration, (b) isoprene emission rate (c) O₃ production rate and (d) net O₃ production rate (production – loss) for the period 2090–2100 for UKESM1. The latitude has been limited to 30°S–13°N to remove Sub-Saharan Africa, which is excluded from the linear model.**

Line 512: Should say which reactions happen faster at higher temperatures.

There are several reactions although the most important seems to be recycling of $NO_2$. Added as shown below.

'

Firstly, the major $O_3$ forming reaction $NO + HO_2 / RO_2$, happens faster at higher temperatures and scaling up $O_3$ production in an area where $O_3$ production is already high will often lead to greater $O_3$ increases than an area with low $O_3$ precursor concentrations (Coates et al. 2016).'

Conclusions: This doesn't discuss why there is an ozone-climate penalty.

Additional comments have been added to the conclusion to cover this point (line 572).

'This is not due to increased fire emissions, but due to the increasing temperature , which speeds up the recycling of NO into NOx and increases decomposition of PAN into NOx in source regions. .'

Line 572: Strictly you have shown that the ozone-climate penalty is in areas of high NOx rather than ozone.

Yes. The wording has been changed, however we maintain that the importance of this result is that the areas of high NOx also tend to have high ozone, and therefore ozone gets worse in areas that already have high ozone.

' We find that while overall $O_3$ concentrations over the tropical land areas are reduced under the climate scenario examined here, climate change could lead to an ozone–climate penalty in areas which have a high background  NOx concentration.  These high-NOx areas already tend to have high $O_3$ concentrations in the absence of climate change (above 40 ppb), with climate change causing a further deterioration in air quality.'

We have also amended the conclusion (line 571):

' Using a multimodel mean of data from three Earth system models, we identify that by 2100, there will be an ozone–climate penalty in high-NOx areas , such as major cities and biomass burning areas (Fig. 4).'

**REVIEWER 2:**

This manuscript presents an interesting work focusing on the climate change effect on near surface ozone over South America and Africa using three ESMs.

It is generally well structured and presented. However I have a number of points that have to be addressed before acceptance. See my analytical comments below:

We thank the reviewer for the detailed and constructive comments. In response, we have made the changes outlined below and slightly restructured / modified Sect. 3.4 and the discussion to describe changes in NOx in further detail.

**Comments**

line 51:It can be also produced from the oxidation of CO in the presence of adequate levels of NOx so that net ozone production (P-L) is positive. Please modify accordingly.

We have now changed this accordingly.

' $O_3$ is a highly oxidising compound, formed in the atmosphere through reaction of volatile organic compounds (VOC) or carbon monoxide (CO) with hydroxyl radicals (OH), and nitrate radicals ($NO_3$) in the presence of nitrogen oxides (NOx).'

line 68: Apart from chemical and biological processes there are also complex dynamical processes such as transport from the stratosphere to troposphere. See for example Morgenstern et al. 2018, Akritidis et al., 2019.

These were assumed to be included in the list under the term 'transport'. For clarity we have modified the sentence.

'Whilst there have been no studies specifically assessing changes in surface $O_3$ due to climate change in the tropics, global studies have suggested that chemical and biological changes in temperature-dependent chemistry, natural emissions of precursors, transport and land surface properties, as well as dynamical changes including circulation changes and transport from the stratosphere may lead to an 'ozone–climate penalty over some continental areas (Jacob & Winner, 2009; Doherty et al., 2013; Zanis et al., 2022).'

line 84: There also soil NOx emissions which are not mentioned (e,g. Romer et al., 2018).

Soils have been added.

'NOx is produced from both natural and anthropogenic sources including soils, lightning, transport and biomass burning.'

line 84: What do the authors mean with severely NOx-limited regions? Please clarify.

The word 'severely' has been removed, as well as further changes in response to comments below.

' Regions are defined as NOx-limited when increasing  VOCs or OH acts to reduce $O_3$ concentrations through oxidation and formation of organic peroxides  (Pacifico et al., 2012). In this NOx-limited case, increasing NOx will lead to greater $O_3$ formation. Conversely, in  VOC-limited regions with sufficient NOx present, increasing NOx concentrations may reduce $O_3$ concentrations by removal of the key $O_3$-forming radicals OH (reaction: OH + $NO_2$ → $HNO_3$).'

line 86-87: Do the authors imply the NO+O3 titration process acting as an ozone sink close to NO emission sources which is common in highly polluted areas being in the VOC limited regime?

Thanks for identifying this unclear section. The paragraph above has been modified and now simply describes the basics of NOx-limited and NOx-saturated regimes.

line 102: Concerning the isoprene chemistry in models, maybe the authors could also refer here more explicitly the uncertainty due to the different model assumptions on the yields of isoprene nitrates and their subsequent NOx-recycling ratios.

Thank you this is a very good idea. The variation in NOx recycling between models is now specifically addressed in the introduction. We have also added a comment on HOx recycling.

'The role of isoprene in the ozone–climate penalty is debated as there is uncertainty about how isoprene emissions will change in the future in response to temperature, $CO_2$ and land-use change (Fu & Liao, 2016; Fu & Tian, 2019) and how to best represent isoprene chemistry in climate models (Weber et al., 2021). Biogenic isoprene emissions increase strongly with temperature and vegetation stress (e.g. Guenther et al., 1993; Niinemets et al., 1999; Unger et al., 2013; Morfopoulos et al., 2021), but very high temperatures or moisture stress may cause 'die-back' of vegetated areas, which would decrease isoprene emissions overall (Sanderson et al., 2003; Cox et al., 2004; Malhi et al., 2009). On the other hand, elevated $CO_2$ concentrations directly inhibit isoprene emission but can indirectly increase emission if this $CO_2$ fertilisation effect results in increased plant productivity (Pacifico et al., 2011; Squire et al., 2014; Hantson et al., 2017). Isoprene, NOx and OH concentrations are influenced by isoprene chemistry. Formation of isoprene nitrates partially recycles NOx, and oxidation of isoprene partially recycles HOx (Bates and Jacob, 2019). Difference between models in their calculation of the yields and recycling rates of these species is likely to affect $O_3$ concentrations.'

line 223: The authors mention that the simulations are fully coupled but the ocean is not coupled since SSTs are prescribed.

True. We use a simulation driven with changing SSTs from the coupled simulation ssp370. This has been corrected.

' The simulations are  global model runs driven with prescribed sea surface temperatures (SSTs) over the period 2015–2100.'

line 230: Apart from the cited reference, this approach has been also applied in Chapter 6 of IPCC AR6 (Szopa et al, 2021).

Thanks. This reference has been included.

line 240: The authors mention that tropics as lying between 40 N and 40 S. The tropics are commonly defined as the area between the Tropic of Cancer (roughly 23.5-degrees North latitude) and the Tropic of Capricorn (roughly 23.5 degrees-South Latitude). In the domain of Figure 1, the subtropics of Northern and Southern Hemisphere are also included. This is a comment to be considered as in several places throughout the manuscript the authors refer to the tropics but the analysis includes also subtropics.

Tropics has been replaced with 'tropics and subtropics' or '40°S–40°N' throughout and a sentence has been added at the start to define that by tropics and subtropics we refer to the latitude band 40°S–40°N.

line 272: Maybe you can also discuss this result in relation to the results of Turnock et al. (2020).

We have added a comment in the discussion although comparison with previous global studies is difficult as they tend to only use the 2 data points that were previously available on TOAR I, so their analysis of South America is limited and Africa is often non-existent.

'We find that the modelled surface $O_3$ bias compared to in situ observations is largest in biomass burning areas, although in South America the models capture the seasonal cycle well (Fig. 1). In situ sites, especially in the DR Congo (Fig. 1e), do not detect the large increases in $O_3$ predicted by models during biomass burning months although observed $O_3$ concentrations are also highest during biomass burning season (Adon et al., 2013). A high positive bias during the dry season has been found in previous studies (e.g. Turnock et al., 2020) although our study has covered several regions that did not previously have available data.'

line 316: Do you have some explanation why NOx surface concentration decreases in UKESM and slightly increase in MRI and GISS in Fig 2b? A discussion with possible reasons is missing.

Adding this has improved the paper significantly. In response to similar comments from reviewer 1, the discussion has been modified to include this, although in this paper we do not conclusively determine the cause of these changes.

In the discussion section (added at line 516):

'As changes in NOx concentrations are shown to be important for changes in $O_3$ production in GISS and MRI, we now discuss the intermodel differences in NOx concentration changes in further detail. GISS and MRI agree that NOx concentrations will increase in high-NOx regions, but disagree on the direction of change in remote regions. UKESM1 predicts a decrease in NOx concentrations in many areas. GISS predicts a decrease in NOx of $2x10^{-11}$ mol mol$^{-1}$ in remote areas (up to 50 %) while MRI predicts an increase of $2x10^{-11}$ to $6x10^{-11}$ mol mol$^{-1}$ in remote regions. This is likely causing the difference in $O_3$ production between GISS and MRI in remote regions. To reduce uncertainty in predictions for $O_3$ concentration

changes due to climate change, further work to constrain future NOx concentration changes is needed.

The NOx concentration change depends on the balance of NOx production and loss terms. A large contributor to increases in NOx concentrations in all models is an increased decomposition of PAN into NOx, which will be largest in source regions. Lightning NOx is also a NOx source, but its influence on surface NOx remains unclear. Although lightning NOx increases in all models during the wet season, but the largest surface NOx and $O_3$ increases occur in the dry season, so the ozone–climate penalty is unlikely to be driven by lightning NOx changes. Nevertheless, the large increase in lightning NOx in MRI may have a role in the increase in surface NOx concentration in MRI, which is larger than the other models, and lightning NOx decreases in the Northern Amazon during the dry season (Fig. S4a) may contribute to the decrease in NOx and $O_3$ production in this region in UKESM1 (Fig S7). A large contributor to the loss term will be reaction with isoprene derivatives, thus increased formation of isoprene nitrates. In both UKESM1 and GISS, isoprene and NOx are anti-correlated in some areas, suggesting isoprene emissions changes have a notable effect on NOx concentrations. For example, GISS predicts large isoprene increases in the remote Amazon, where the major NOx decreases occur, and UKESM1 shows a small area of increased NOx concentrations downwind of the Northern Amazon, where isoprene decreases. As MRI prescribes isoprene as a climatology, there is will be no significant change to NOx loss via organic nitrate formation and this is likely a reason why NOx increases over most areas of land. The reasons why NOx loss dominates in UKESM1 whereas GISS shows a net NOx increase is likelyrequires further understanding of due to individual model details such as isoprene nitrate yield and NOx recycling frequency. The sensitivity of NOx concentration to changes in lightning, PAN and isoprene in each model would be useful to explore in further work. Further studies could also explore some temperature-sensitive sources of NOx that were not included in the simulations such as soil NOx emissions and changes in wildfire frequency.'

line 316: There is no discussion of OH changes in Fig. 2c. Could you discuss why MRI and UKESM show an increase while GISS shows a decrease? Theoretically, increases in methane, CO and NMVOCs reduce OH while increases in water vapour and temperature, incoming solar radiation, NOx and tropospheric ozone enhance OH.

As OH is a major radical in many reactions, we feel it is beyond the scope of this study to determine the complete reasons why OH concentrations change. However, figures shown in the supplementary identify changes in isoprene as an important factor in reducing OH concentrations in GISS (and to some extent in UKESM1) and further text has been added to the main paper.

'Hydroxyl radical concentration (OH) determines the oxidising capacity of the atmosphere and affects rates of reaction such as VOC oxidation and ozone destruction. Increased temperatures will increase atmospheric water vapour and OH production, however OH concentrations decrease in the GISS model when climate change is included. A portion of this decrease can be attributed to an increase in isoprene emissions, which is much larger in GISS than UKESM1 (Fig. S2)'

line 331: Please discuss also how the models deal with the soil NOx emissions in the simulations.

Now included. Soil NOx is calculated individually for each model but does not respond to changes in climate in any model.

'NOx emissions, including biomass burning emissions, are prescribed based on the SSP3-7.0 scenario but lightning NOx and soil NOx differs between the models based on the chosen parameterisation of individual models. Compared to the present-day, NOx emissions in biomass burning areas decrease in Africa to follow projected trends, but do not change in South America. NOx emissions increase in cities and Nigeria especially has major growth in urban areas. Compared to the scenario without climate change, total lightning NOx emissions increase in all models, and the locations of the increases vary in latitude to follow the wet season (Fig. S4). MRI predicts much larger increases than GISS and UKESM1, and UKESM1 shows a decrease in lightning NOx over the Amazon basin in DJF (Fig. S4a) although the net effect over all seasons is positive. PAN decreases in all models (-94 ppt, -61 ppt, -30 ppt for UKESM1, GISS and MRI respectively) likely due to increased thermal decomposition. In GISS and UKESM1, the increase in isoprene emissions can increase removal of es NOx via formation of isoprene nitrates. Soil NOx does not change in response to climate change in any model.  '

lines 387-389: It is rather confusing when discussing megacities and urban scale in model results with much coarser resolution. The current datasets produced for CMIP6 were produced at 0.5â˜ (historical anthropogenic and future) (Feng et al., 2020). Capturing the distinctions between urban and rural emissions, and the finer distinctions in between, is an ongoing challenge for emission inventories for global datasets. I think that the discussion should rather point that areas with ozone penalty identified in this study include a number of highly populated regions and megacities.

A very good point and language has been adapted in this paragraph accordingly.

'  Grid cells which include highly populated regions and megacities  in $O_3$ concentration in all months,  and an average ozone–climate penalty of 3 ppb in the yearly average. In particular, there is an ozone–climate penalty of 3 ppb that shows limited seasonal variation in grid cell containing the megacities in Nigeria (Lagos), Brazil (São Paulo, Rio de Janeiro) and Colombia (Bogotá, Medellín).  This penalty is robust over Southeast Brazil in all seasons (Fig. 4).'

line 390: The authors introduce in Section 3.4, chemical production and loss terms but they have to describe how these terms are calculated (simply discuss the model diagnostics for these terms).

These have now been included in the methods section (line 267-269).

'O$_3$ chemical production is defined as O$_3$ produced from the reaction NO+ RO$_2$ / HO$_2$, O$_3$ chemical loss is the sum of (i) O(1D)+H$_2$O; (ii) O$_3$+HO$_2$; (iii) O$_3$+OH; (iv) O$_3$+alkenes (e.g. isoprene). '

lines 397-398: There is missing discussion on the reasons for the decrease of dry deposition of ozone due to climate change. Is this a robust signal? There are different physical and biophysical processes affecting dry deposition of ozone on which climate change may induce opposite sign of change. To what level these processes are parameterized in the model simulations is crucial for the understanding.

Indeed, a discussion of dry deposition was missing. It is now present but limited as further details on the parameterisations of the land surface models are not available.

In Sect. 3.4 (line 414):

'Conversely, the deposition rate decreases, presumably because the increased temperatures and lower relative humidity cause stomatal closure. Dry deposition varies between each model depending on stomatal response to temperature changes and boundary layer resistance changes (Fig. S10). In UKESM1, the increase in CO$_2$ also reduces stomatal conductance. UKESM1 shows a large decrease in deposition rate over the central Amazon, whereas MRI shows very little change regionally. '

In discussion:

'The decrease in deposition rate is controlled by UKESM1 and GISS (MRI showed very little change), but there was spatial variation in the magnitude of the change. This could be due to changes in meteorology between models (such as temperature and precipitation), as well as model differences (UKESM1 includes CO$_2$ inhibition). Feedbacks such as O$_3$ damage to vegetation were not considered in any model (Pacifico et al., 2015) but may be a useful addition to future simulations.'

lines 412-413: This is a comment linked to my previous comment on dry deposition of ozone. Please check to what level biosphere-atmosphere interactions are taken into account in the current simulations (negative and positive feedbacks on ozone through stomatal upatake and deposition velocity reduction). Mind also the links of dry deposition velocity with the boundary layer changes under a warmer climate.

To our best efforts we have tried to show variation in dry deposition between models, including a plot of dry deposition rate in the supplementary (Fig. S10). However, a detailed breakdown of changes in BL resistance and stomatal conductance are not available as output. Since the contribution from dry deposition is small compared to ozone production, attribution of changes in dry deposition in each model to changes in the land surface is beyond the scope of the paper.

[Figure]

**Figure S10: The average change due to climate change in ozone dry deposition rate for the period 2090 - 2100 for (a) UKESM1, (b) GISS and (c) MRI.**

lines 437-438: Could you commend on the possible reasons that determine the net ozone production increase for UKESM? Despite NOx decrease in many regions due to natural emission changes and chemistry under climate change, net ozone production is positive. So it is essential the high NOx levels at polluted regions (NOx regional levels) which play a key role for an increase in net ozone production rate in a warmer and more humid environment. On top of that there also isoprene emission increases which may have an impact on ozone and it could useful to discuss their contribution on net ozone production rates (in which regions have a positive or negative impact and if this is related to the regions NOx levels).

This is a key point that was lacking in the paper.

We have now made clear reference to temperature as an additional variable that controls ozone production, and attribute the increase in ozone production in UKESM1 largely to the increase in reaction rate of NO + HO2 / RO2 at higher temperatures.

Edits to line 510:

[revised manuscript text omitted]

line 502: It is rather misleading when the authors state that "climate change could lead to an ozone–climate penalty in areas which already have a high background O3 concentration". The ozone climate penalty is linked to the regional NOx levels in a highly polluted region where net ozone production rates increase in warmer and

more humid environment (even if natural NOx levels are slightly reduced due to climate change).

Agreed. The text has been revised, although we feel the importance of the ozone-climate penalty in relation to the background ozone concentration is still worth mentioning (but we try not to imply a cause and effect relationship).

' We find that while overall $O_3$ concentrations over the tropical land areas are reduced under the climate scenario examined here, climate change could lead to an ozone–climate penalty in areas which have a high background  NOx concentration.  These high-NOx areas already tend to have high $O_3$ concentrations in the absence of climate change (above 40 ppb), with climate change causing a further deterioration in air quality.'

Also the conclusion (line 571):

' Using a multimodel mean of data from three Earth system models, we identify that by 2100, there will be an ozone–climate penalty in high-NOx areas, such as major cities and biomass burning areas (Fig. 4).'

Figure 6: A letter should be assigned in each one of the 12 sub-figures of the panel.

Done.

**Minor technical comments**

lines 26-27: The whole sentence needs rephrasing as it does not read well.

Agreed, and the sentence has been modified accordingly:

'

In order to quantify changes due to climate change, we evaluate the difference between simulations including climate change and simulations with a fixed present-day climate.'

line 30: Please add a "comma" after "on average".

lines 211-212: The sentence needs rephrasing as it does not read well.

Agreed, and the sentence has been modified accordingly:

'

To produce a model seasonal cycle, the monthly mean $O_3$ concentration was calculated using CMIP6 historical simulations for the period 1990–2014. This was done for each grid cell that contained an observation site. '

line 226: Please replace "are" with "and".

line 271 : Please replace "the" with "that".

line 291: I would suggest "range between" instead of "fall within".

line 376: It is Fig. 4c instead of Fig.4d.

line 377: It is Fig. 4d instead of Fig.4c.

line 382: I would suggest "larger" than "much more extreme".

We thank the reviewer for identifying these mistakes. All have been corrected.

---

## Author Comment (AC2)

COMMENT FROM REVIEWER (page 4, line 15 of previous author comments):

Using monthly values in figure 7 might lead to spurious correlations. There will be strongly seasonal variations in isoprene and NOx, and also strong seasonal variations in meteorology (wet vs dry). Some of the correlations in figure 7 might be due to the meteorology -i.e. isoprene emission changes might be stronger in the dry season where the meteorological impacts on ozone might also be more positive if dry gets drier.

The author response remains unchanged, however the authors would like to update the quoted changes to the manuscript (page 5, line 8 of the previous author comments). Amendments are shown in **bold** below.

PREVIOUS TEXT:

'The change in $O_3$ production rate will be further affected by meteorological changes, temperature in particular. This is the reason that $O_3$ production increases in UKESM1 and MRI even in the absence of changes in NOx and isoprene (the intercepts of the linear model are 19 % and 5 % respectively) and $O_3$ production increases in areas showing decreasing NOx concentrations in UKESM1. Since the temperature change varies seasonally and regionally, with dry seasons experiencing the largest increase in temperature, some of the changes in $O_3$ production in Fig. 7 may be driven by temperature rather than NOx or isoprene changes. If isoprene/NOx and $O_3$ production are both influenced by the underlying meteorology, the identified correlations may be due to meteorology rather than the chemical species changes. We verify that percentage NOx change is not related to temperature in any of the models and that percentage isoprene change is not related to temperature in UKESM1 (not shown), which indicates the identified correlations are related to chemical species changes not meteorological variation, although this cannot be entirely ruled out.'

THE MANUSCRIPT NOW READS:

'The change in $O_3$ production rate will be further affected by meteorological changes, temperature in particular. This is the reason that $O_3$ production increases in UKESM1 and MRI even in the absence of changes in NOx and isoprene (the intercepts of the linear model are 19 % and 5 % respectively) and $O_3$ production increases in areas showing decreasing NOx concentrations in UKESM1. Since the temperature change varies seasonally and regionally, with dry seasons experiencing the largest increase in temperature, some of the changes in $O_3$ production in Fig. 7 may be driven by temperature rather than NOx or isoprene changes. If isoprene/NOx and $O_3$ production are both influenced by the underlying meteorology, the identified correlations may be due to meteorology rather than the chemical species changes. **We verify that the monthly mean temperature change in each gridcell is not significantly correlated with percentage NOx change in any model, nor percentage isoprene change in UKESM1 (not shown). Therefore, NOx and isoprene changes are likely controlled by many processes in addition to temperature, including background chemistry and emissions for NOx, and vegetation type and cover for isoprene, as well as other meteorological variables. This indicates that the identified correlations between NOx and $O_3$ production are unlikely to be the result of a spurious relationship driven by temperature, although it is still possible that the strength of the correlations may be inflated by confounding meteorological variables.**'

---

## Author Response (AR2)

Author response:

We thank the editor for their comments, which have helped us to improve our manuscript. All comments have been addressed. The editor comment is shown in black, our reply is shown in blue and extracts from the revised manuscript text are shown in 'commas', with altered text in red and deleted text with .

(1) A very technical suggestion, but your statement: "Regions are defined as NOx-limited when increasing VOCs or OH acts to reduce O3 concentrations..." seems a bit too categorical in my view. NOx-limited regimes can certainly include situations O3 production is not sensitive to perturbations in VOC abundance (see for example Figure 12-4 in Jacob (1999), Introduction to Atmospheric Chemistry). I would suggest wording along the lines of:

"In NOx-limited regimes where O3 production is proportional to NOx concentrations, increasing VOCs or OH can also act to reduce O3 concentrations..."

We thank the editor for this suggestion and have modified the sentence:

'In NOx-limited regimes where $O_3$ production is proportional to NOx concentrations, increasing VOCs or OH can also act to  reduce $O_3$ concentrations through oxidation and formation of organic peroxides (Pacifico et al., 2012). In this NOx-limited case, increasing NOx will lead to greater $O_3$ formation.'

(2) One of the reviewers asks, "Please discuss also how the models deal with the soil NOx emissions in the simulations.". In your response you state:

"NOx emissions, including biomass burning emissions, are prescribed based on the SSP3-7.0 scenario but lightning NOx and soil NOx differs between the models based on the chosen parameterisation of individual models"

Should this sentence be, "Anthropogenic NOx emissions, including biomass burning emissions are prescribed based on..."? Otherwise it's confusing since you first state that NOx emissions are prescribed, but then state certain sources that are not. Please clarify.

We thank the editor for identifying this mistake and the sentence has been corrected:

'Anthropogenic NOx emissions, including biomass burning emissions, are prescribed based on the SSP3-7.0 scenario, soil NOx is prescribed by each model and  lightning NOx  differs between the models based on the chosen parameterisation of individual models. Compared to the present-day, NOx emissions in biomass burning areas decrease in Africa to follow projected trends, but do not change in South America. NOx emissions increase in cities and Nigeria especially has major growth in urban areas. Compared to the scenario without climate change, total lightning NOx emissions increase in all models, and the increases occur during the wet season (Fig. S4). MRI predicts much larger increases than GISS and UKESM1, and UKESM1 shows a decrease in lightning NOx over the Amazon basin in December–February (Fig. S4a) although the net effect over all seasons is positive. Peroxyacetyl nitrate (PAN) decreases in all models (−94 ppt, −61 ppt, −30 ppt for UKESM1, GISS and MRI respectively) due to increased thermal decomposition. In GISS and UKESM1, the increase in isoprene emissions can increase removal of NOx via formation of isoprene nitrates. '

On the topic of this reviewer's same comment, in your Model Descriptions Section 2.1, I suggest that you include very brief descriptions of the isoprene and soil NOx parameterization schemes used in the models where these emissions are calculated interactively (in addition to just including the relevant citations). In this section, I would also encourage you to be explicitly clear as to what natural emissions will (and will not) respond to the climate component of the simulations (for example, I would almost recommend a short table). Furthermore, when these emissions are prescribed, please consider including the time period over which these emissions are prescribed (e.g., "based on a climatology from Year X to Year Y").

A table has been included and greater detail has been added to the descriptions for each model including the years of the climatologies where available.

**'2.1 Model descriptions**

[revised manuscript text omitted]